# Analysis of Vegetative Cover Vulnerability in Rohingya Refugee Camps of Bangladesh Utilizing Landsat and Per Capita Greening Area (PCGA) Datasets

Md Fazlul Karim [1,2,3,4,5,6] and Xiang Zhang [1,7,*]

1   School of Geospatial Engineering and Science, Sun Yat-sen University, Guangzhou 510275, China; f.karim7631@nirapad.org.bd
2   State Key Laboratory of Information Engineering in Surveying, Mapping and Remote Sensing, Wuhan University, 129 Luoyu Road, Wuhan 430079, China
3   School of Resource and Environmental Sciences, Wuhan University, 129 Luoyu Road, Wuhan 430079, China
4   Department of Geography and Environment, Jagannath University, 9-10 Chittaranjan Avenue, Dhaka 1100, Bangladesh
5   Shahidul Consultant Ltd., 66/D Indira Road, Farmgate, Dhaka 1215, Bangladesh
6   Network for Information, Response and Preparedness Activities on Disaster (NIRAPAD), 4/16 (1st Floor), Humayun Road, Block-B, Mohammadpur, Dhaka 1207, Bangladesh
7   Southern Marine Science and Engineering Guangdong Laboratory, Zhuhai 519082, China
*   Correspondence: zhangx795@mail.sysu.edu.cn

**Abstract:** The vegetative cover in and surrounding the Rohingya refugee camps in Ukhiya-Teknaf is highly vulnerable since millions of refugees moved into the area, which led to severe environmental degradation. In this research, we used a supervised image classification technique to quantify the vegetative cover changes both in Ukhiya-Teknaf and thirty-four refugee camps in three time-steps: one pre-refugee crisis (January 2017), and two post-refugee crisis (March 2018, and February 2019), in order to identify the factors behind the decline in vegetative cover. The vegetative cover vulnerability of the thirty-four refugee camps was assessed using the Per Capita Greening Area (PCGA) datasets and K-means classification techniques. The satellite-based monitoring result affirms a massive loss of vegetative cover, approximately 5482.2 hectares (14%), in Ukhiya-Teknaf and 1502.56 hectares (79.57%) among the thirty-four refugee camps, between 2017 and 2019. K-means classification revealed that the vegetative cover in about 82% of the refugee camps is highly vulnerable. In the end, a recommendation as to establishing the studied region as an ecological park is proposed and some guidelines discussed. This could protect and reserve forests from further deforestation in the area, and foster future discussion among policymakers and researchers.

**Keywords:** Rohingya refugee crisis; vegetative cover vulnerability analysis; Landsat; supervised image classification; per capita greening area (PCGA) datasets; K-means classification; spatiotemporal change analysis

## 1. Introduction

The local inhabitants, previously settled Rohingya refugees, and different national and international non-governmental organizations (NGOs) have built approximately 209,891 households for nearly a million Rohingya refugees in the Ukhiya-Teknaf upazila of Bangladesh [1] since the commencement of the Rohingya refugee crisis on 25 August 2017 [2–8]. This exodus is regarded as one of the fastest developing refugee crises since World War II [9–11]. Rohingya minorities are one of the most widely victimized refugees in the world at the moment; the government of Bangladesh refers to Rohingya refugees as "Forcibly Displaced Myanmar Nationals" [12–14]. In response, the government of Bangladesh allocated 1942.49 hectares of mostly hilly-forested land for refugee settlement in September 2017, which quickly became overpopulated [6]. As of April 2019, more than

610,372 refugees were accommodated in that camp, and the number is growing each day (see Figure 1).

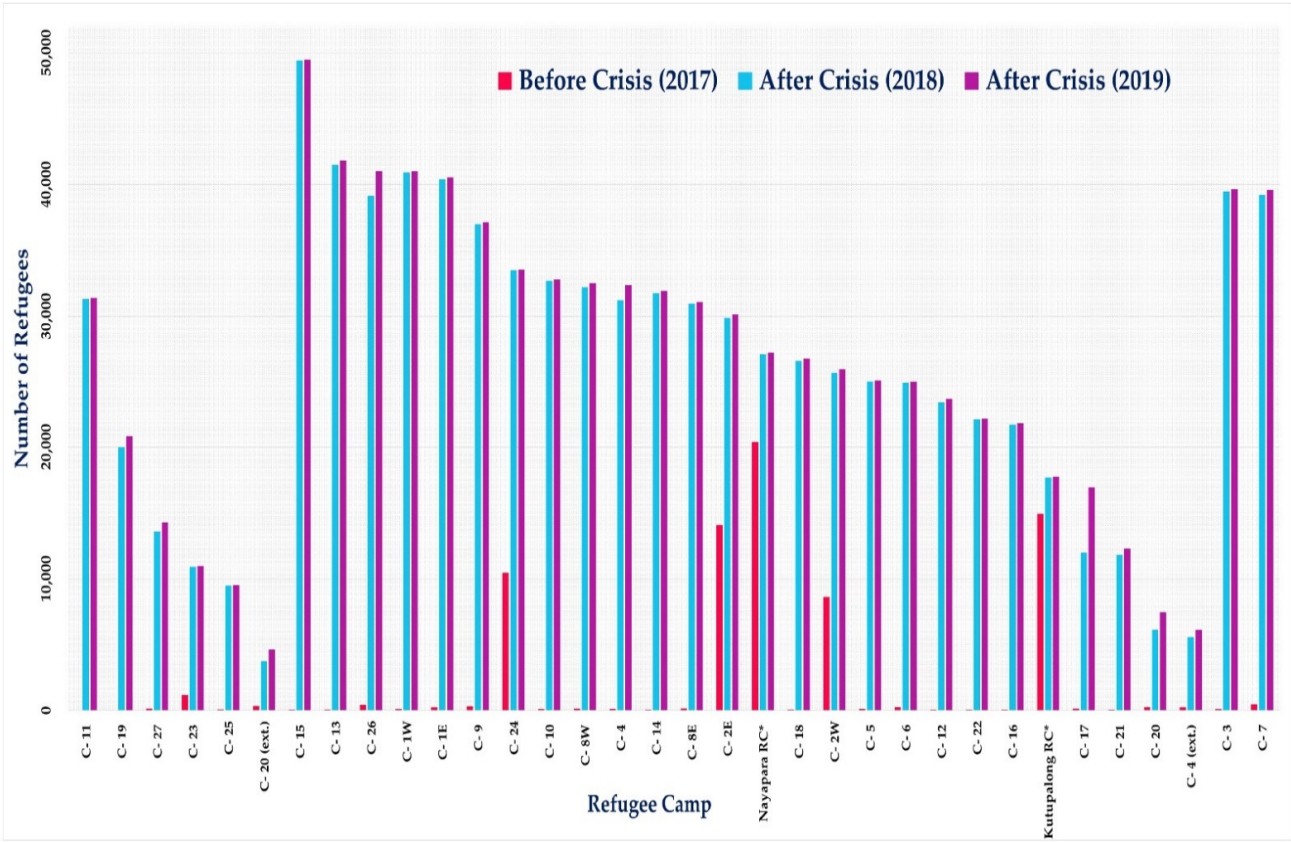

**Figure 1.** Camp-wise Rohingya refugee changes, pre and post refugee crisis (between 2017 and 2019), among thirty-four refugee camps in Ukhiya-Teknaf. Here, C = camp; RC* = registered camp; Ext. = extension. (Data source: [10]).

The vegetative cover of Ukhiya-Teknaf is under severe pressure due to the unregulated expansion of refugee settlements [2,4,9,15]. The rapid conversion of lands with vegetative cover into refugee settlements is the principal agent of environmental degradation and has caused the most significant large-scale land cover changes in Ukhiya-Teknaf in recent times [14]. Hassan and Smith (2018) quantified the territorial expansion of refugee settlements in Teknaf, finding that the refugee settlement increased from 175 hectares to 1530 hectares between 2016 and 2017 [2]. Imtiaz (2018) assessed the impact on the vegetative cover by displaced refugees on the Teknaf peninsula, estimating a total vegetative cover loss of 1284.48 hectares between 2014 and 2017 [4]. Labib et al. (2018) determined vegetative cover and estimated carbon emission losses in the Kutupalong-Balukhali expansion site: 572 hectares of vegetative cover were deforested to set up camps between 2016 and 2017, which accounted for an approximate loss of 365,288 Great British Pounds (GBP) per year [15].

There are thirty-four Rohingya refugee camps settled over Ukhiya-Teknaf and they are equally responsible for the overall vegetative cover loss, but Imtiaz (2018) assessed the vegetative cover changes only in Teknaf, and Labib et al. (2018) in Kutupalong-Balukhali Rohingya refugee expansion sites [4,15]. With the intention of calculating the vegetative cover loss and territorial expansion of the Rohingya refugee settlement in Teknaf, Hasan et al. (2018) created multiple ring buffer zones centered on the three pre-existing refugee camps to narrow down their analysis [2]. However, previous studies did not attempt to define the actual vegetative cover changes among all thirty-four refugee camps, which is crucial to fathom the influence of Rohingya refugees on vegetative cover loss and eventually identify refugee camps with highly vulnerable vegetative cover. Furthermore,

the introduction of the per capita greening area (PCGA) dataset in this research deepens our understanding of the vegetative cover capacity changes at each of the thirty-four refugee camps. PCGA datasets are the ratio between each of the thirty-four refugee camps vegetative cover and the number of refugees in 2017, 2018, and 2019. The combination of Landsat satellite data and the PCGA dataset is a novel approach to investigate the possible vegetative cover vulnerability of the refugee camps in Ukhiya-Teknaf.

The ultimate goal of this research is to identify the critical factors behind the declining vegetative cover in the Ukhiya-Teknaf area of Bangladesh, as well as quantify Rohingya refugee camps with highly vulnerable vegetation cover using Remote Sensing (RS), Geographical Information System (GIS), and Machine Learning (ML) techniques. The results of this study are indispensable and should be able to help the bodies responsible for international human rights, refugee welfare, policymakers, and all the national and international organizations to comprehend the challenges (e.g., environmental degradation) facing the Bangladesh government due to the refugee issues. This research might be instrumental to researchers in multiple subjects (e.g., refugee studies, political geography, international relationships, migration, environmental sciences, and so on) and could be influential in taking initiatives to reduce deforestation and forest degradation activities.

The objectives of this research are twofold. First, we map and characterize temporal changes of vegetative cover in three phases between 2017 and 2019, more specifically pre and post influx of refugees in both Ukhiya-Teknaf and the thirty-four refugee camps, in order to identify the critical factors behind the declining vegetative cover. Second, we quantify the per capital greening area changes at each of the camps due to the sudden influx of refugees and identify Rohingya refugee camps with highly vulnerable vegetation cover over the study period, with the intention of monitoring the vegetative cover changes in the Ukhiya-Teknaf area of Bangladesh.

## 2. Related Work

Land use/land cover change (LULCC) shows the existing interactions between the physical and human environment [16]. The foremost benefits of using satellite data and GIS techniques for analysis of land cover change are its timely, cost-effective, and labor-saving felicity. Ali et al. (2018), Ayele et al. (2018), and Rimal et al. (2018) performed maximum likelihood (ML) classification and Landsat for land cover classification [17–19]. Hassan et al. (2018) and Phiri et al. (2018) used random forest (RF) classification for land cover change analysis [2,20,21]. Yoo et al. (2019) used the convolutional neural network (CNN) for climate zone classification [22]. Several studies already revealed that the Support Vector Machine (SVM) supervised image classifier more precisely classified than the other classification techniques. Topaloglu et al. (2016) applied maximum likelihood (ML) and SVM to classify eight different land categories in Istanbul in Turkey to compare classification accuracies of classified maps, and stated that SVM produces better results compared to ML [23]. Yousefi et al. (2015) compared six supervised classification algorithms, namely minimum distance of mean (MDM), mahalanobis distance (MD), ML, artificial neural network (ANN), spectral angle mapper (SAM), and SVM in terms of land-use mapping in Iran, and they show that SVM outperformed the others [24]. The image classification algorithms and high-resolution satellite imagery makes data mining and monitoring of a broad range of target features on the ground relatively trouble-free.

Specifically, SVM is an efficient learning algorithm for remote sensing classification applications, for instance, vegetative cover monitoring [25–27]. SVM classification is applied broadly to achieve different research objectives due to its high classification accuracy and ability to handle complex relations among variables [28]. Although LULCC analysis depicts the relationship between the human and physical environment, the use of diverse datasets (e.g., satellite data and socioeconomic data) with clustering algorithms can produce detailed information and facts. For example, hierarchical clustering [29], K-means [30–32], and Gaussian mixture model [33] are a few benchmark clustering techniques for change analysis. The hierarchical clustering method is preferable with larger datasets, but k-means

clustering performs better both with the large and medium datasets [34]. Karim et al. (2019) used k-means classification to quantify the changing pattern of shrimp yield in three coastal districts of Bangladesh from 2002 to 2017 [30]. For water quality analysis, Zou et al. (2015) utilized the k-means classification technique and took the Heihe River in China as a study area [32]. Agarwal et al. (2013) intended to specify the crime trends of England and Wales and used the k-means method for crime analysis [31].

## 3. Materials and Methods

### 3.1. Study Area

The study area is Ukhiya-Teknaf, two adjoining sub-districts of Cox's Bazar district, located in the southernmost part of Bangladesh, shown in Figure 2, and ranging between 20°43′0″ N and 21°18′0″ N latitude and 92°4′0″ E and 92°20′0″ E longitude. The fastest-growing refugee camps in the world are located in Ukhiya-Teknaf, settling in different positions both in the pre-established and spontaneous new refugee camps that have vaulted up in the region since August 2017 [35]. Ukhiya-Teknaf has 147 villages grouped into 11 clusters (called unions), and 1 municipality (locally known as Paurashava) that totally covers 557 square km (55,700 hectares). Since the crisis broke out, there have been thirty-four refugee camps with a total cumulative refugee population of approximately one million by the time of this research. Ukhiya-Teknaf has a dynamic waterway system with two main water channels—the Naf River and the Reju Canal.

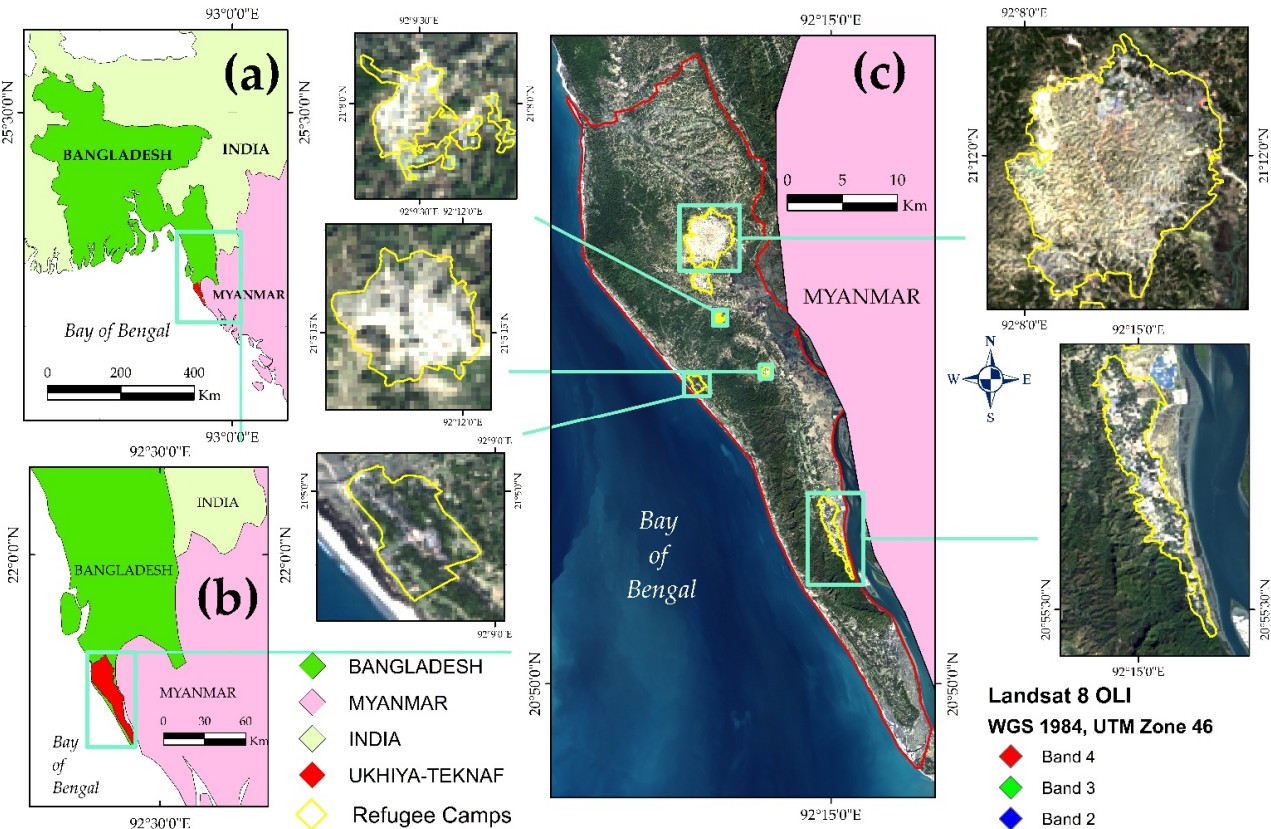

**Figure 2.** A schematic framework of Ukhiya-Teknaf, study area. (**a**) shows Bangladesh bordered by India, Myanmar, and the Bay of Bengal. (**b**) displays the Ukhiya-Teknaf sub-district, represented as red color, bounded by the Ramu sub-district on the north, Arakan state of Myanmar and Naikhongchori sub-district of Bangladesh on the east, and the Bay of Bengal on the west and south corner. These two-adjoining sub-districts are the focus of our analysis of temporal vegetative cover monitoring before and after the Rohingya refugee crisis and eventually identifies the influencing factors behind the declining vegetative cover. (**c**) shows a natural-color composite image of the study area, Ukhiya-Teknaf, with a band combination of 4, 3, and 2 showing all the existing Rohingya refugee camps dated 4 February 2019. These camps are the focus of our identification and analysis of refugee camps with highly vulnerable vegetative cover.

The study area, Ukhiya-Teknaf, is situated in the subtropical monsoon climatic region, with broad seasonal distinctions in rainfall, high temperature, and humidity. Ukhiya-Teknaf experienced three distinct seasons: a dry and cold winter (October–March), a hot summer (March–June), and rainy monsoon (June–October). The monthly average dry bulb temperature from 1975 to 2016 was 25.90 °C/month, and the warmest and coldest months are May (32.2 °C) and January (14.9 °C). The annual average rainfall from 1977 to 2016 was 4067.99 mm/year, the highest rain occurred in July (1029 mm), and the least rain occurred in January (2 mm).

Ukhiya-Teknaf is geographically located in the coastal area and often falls victim to cyclones, sea storms, and tidal bore [36]. The maximum extension of the study area is about 65 km in the north-south, and 10 km in the east-west direction. The study area displays distinct physiographic features such as piedmont plains, rugged rocky hills, and an unbroken 120 km (75 miles) long line of sandy beaches straightening to Cox's Bazar throughout the Bay of Bengal, reportedly the second-largest uninterrupted sea beach in the world after Brazil's Praia do Cassino Sea beach. The sea beach is backed by gently sloping foothills and generally occupied by human activities. The sandy soil, saline water, and a vast area of dense vegetation is the main barrier in the study area for traditional agricultural practices, mainly rice.

There are two registered refugee camps, named Kutupalong RC and Nayapara RC, and thirty-two other refugee camps located in the study area [9]. By the time of this study, the largest refugee camp by refugees was Camp 15, hosting approximately 49,468 refugees. The oldest and second-largest of all the camps is Nayapara RC, housing approximately 37,000 refugees. Camp 20 (extension), located along the north of Camp 15, hosts the fewest refugees, approximately 4630 (see Figure 3).

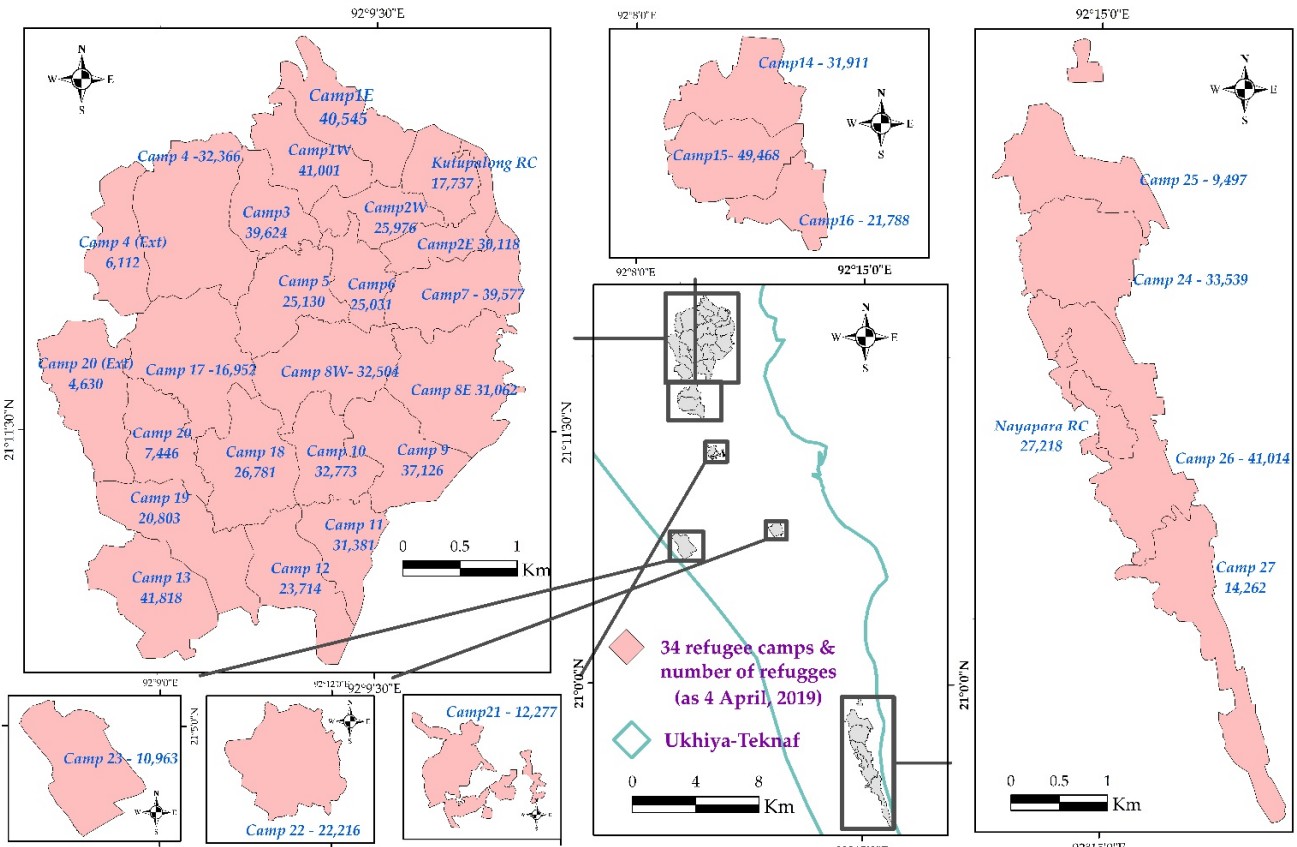

**Figure 3.** Location and representational map of all the thirty-four Rohingya refugee camps and the number of refugees in each camp, as of 4 April 2019 (data source: https://www.unhcr.org/ (accessed on 20 April 2019)).

The subtropical monsoon climatic pattern, along with the physiographic features of Ukhiya-Teknaf, contributes to the development of dense forestland. There are 11,615 hectares of forestland in Ukhiya-Teknaf that are set as a wildlife asylum. This vast area harbors many endangered species such as shoreline and offshore birds, wild Asian elephants, and so forth. About 58.37% of the study area is covered with dense forests, sparse vegetation, and so forth. However, this has declined over time. The cause of this decline is uncertain, but large-scale anthropogenic activities in past years might have triggered substantial loss of vegetative cover [2,4,15,37,38]. The data selection, collection criteria, and satellite imagery processing are elaborated in the next section.

### 3.2. Data Collection and Satellite Image Processing

Diverse demographic, satellite, and geospatial data (presented in Table 1) were utilized to accomplish the goals of this research. Three sets of time period multispectral Landsat 8 OLI/TIRS satellite image scenes were acquired for monitoring vegetative cover. The image scenes at 30 m ground resolution were collected from the U.S. Geological Survey (USGS) website. To obtain cloud-free satellite scenes of Ukhiya-Teknaf, images from the winter seasons (December–February) were selected: the cloud cover was about 0%–10%. After examining all nine images visually, three images were selected for SVM supervised image classification. In this research, ENVI 5.3 was used for atmospheric correction to conduct Fast Line-of-sight Atmospheric Analysis of Hypercubes (FLAASH), for radiometric calibration, and for band combination. The minimum noise fraction (MNF) wizard was used in this study in order to segregate noise from the data, and to reduce the computational requirements for the subsequent processing. Additionally, to prevent spatial referencing problems, all images were registered in the same projection (UTM, WGS 84, zone 46N). In addition, for photo-interpretation of the Landsat images, two historical Google earth scenes of February 2017 and 13 February 2018, and field survey data were used for identifying land cover features, tracing new and old refugee settlements, generating training samples for land cover classification, and in accuracy assessment.

**Table 1.** Representations of the selected satellite, geospatial, and demographic data. OLI, Operational Land Imager; TIRS, Thermal Infrared Sensor; USGS, U.S. Geological Survey.

| Data | Acquired Date/Year | Producer |
| --- | --- | --- |
| Landsat 8 (OLI/TRIS) | 27 January 2017<br>3 January 2018<br>4 February 2019 | USGS global land cover Facilities (http://glovis.usgs.gov/) accessed date 5 March 2019 |
| Geospatial data (Administrative boundary of Ukhiya-Teknaf, Camp location) | 2019 | Humanitarian Data Exchange (HDX) (https://data.humdata.org/) accessed date 15 April 2019 |
| Google Earth Historical Imageries | 13 February 2017,<br>13 February 2018 | Digital Globe |
| Refugee Counts | ISCG, UNHCR,2017, 2018, and 2019 | United Nation (UN) (https://www.unhcr.org/) accessed date 20 April 2019 |
| Vegetation Area Counts | Classified Image of 2017, 2018, and 2019 | Support Vector Machine (SVM) Supervised Classification |

Demographic data for Ukhiya-Teknaf were required to understand the impact of refugee activities on the surrounding environment. The incoming refugees appearing in both the registered and non-registered population data were assembled from various national and international organizations such as Bangladesh Bureau of Statistics (BBS), Inter-Sector Coordination Group (ISCG), and United Nation High Commissioner for Refugees (UNHCR). The vegetative cover data of each of the thirty-four refugee camps was calculated through the classified images of 2017, 2018, and 2019. The refugee population data and

vegetative area data for the thirty-four camps were combined to create the Per Capital Greening Area (PCGA) dataset (see Table 1).

### 3.3. Image Classification

The decisive goal of this research was to detect the critical factors behind the diminishing vegetative cover in Ukhiya-Teknaf. Hardy et al.'s (1976) first-order hierarchical classification system was partly adopted for land cover feature selection [39]. Initially, nine land cover categories (vegetative, agricultural land, aquaculture land, settlement, water-body, arable land, bare land, sandy area, and tidal mudflat) were produced using the professional familiarity of the study area, field survey data, and observations of Google Earth historical images of 13 February 2017, and 13 February 2018, and photo interpretation to identify and confirm the diverse land cover features. The training areas were produced utilizing polygon vectors for individual features based on the spectral reflectance wavelength, presented in Figure 4 and Table 2.

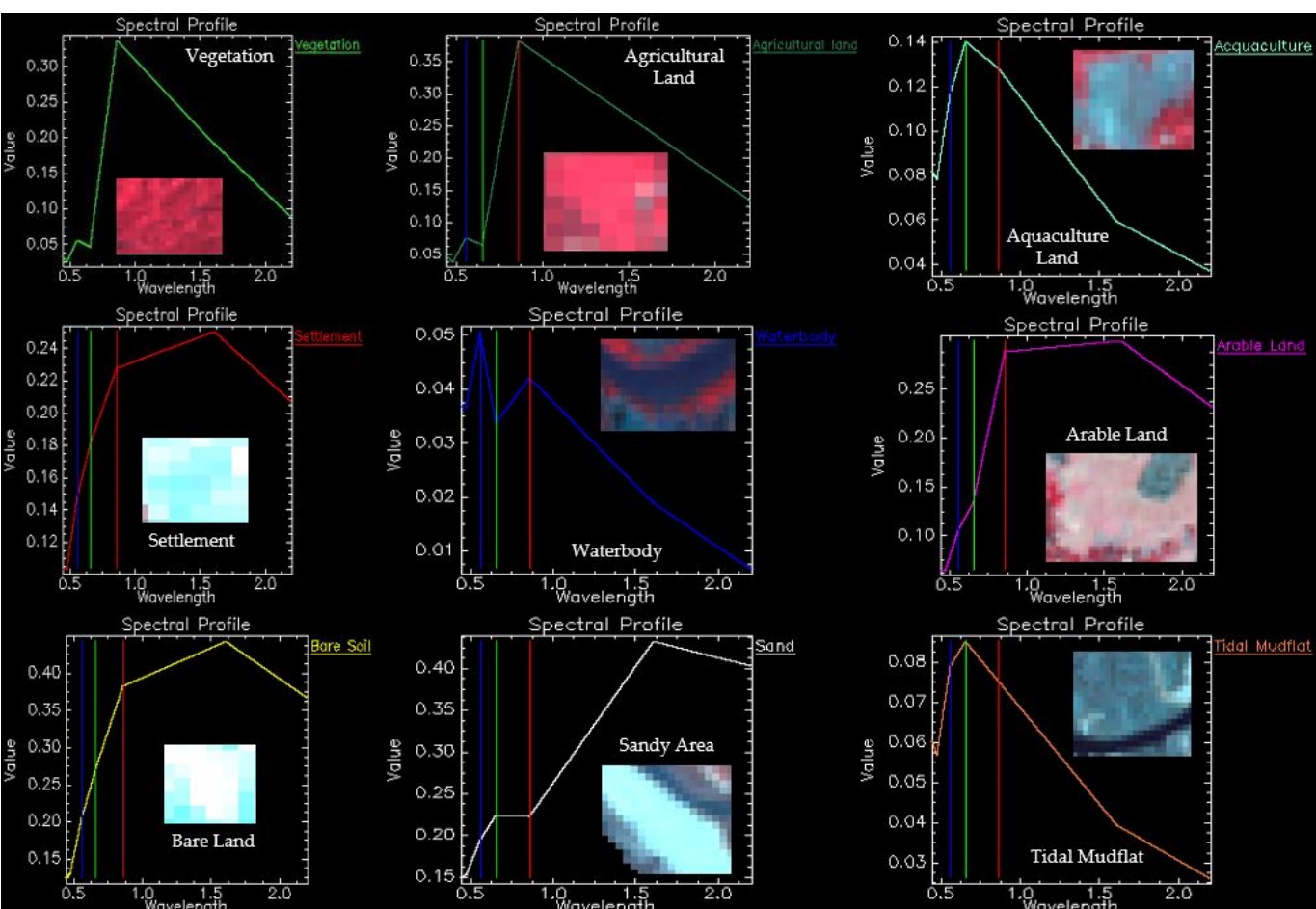

**Figure 4.** Selected training features spectral reflectance wavelength. Each color represents the mean reflectance wavelength value of different land cover classes. In here, light green = vegetation, dark green = agricultural land, cyan = aquaculture land, red = settlement, blue = waterbody, pink = arable land, yellow = bare land, white = sandy area, and coral = tidal mudflat (based on [30]).



**Table 2.** Land use/land cover (LULC) classes delineated based on supervised classification (based on [2,39,40]).

| N | Land Cover Type | Description |
|---|---|---|
| 1 | Vegetation | Scattered forest, mixed forest, sparse low-density forest, degraded forest, the mix of trees and other natural grass covers, homestead vegetation. |
| 2 | Agricultural land | Wet and dry crop fields, paddy fields, fallow lands. |
| 3 | Aquaculture land | Marine aquaculture, brackish water shrimp farming area containing saline. |
| 4 | Settlement | Isolated and clustered small and large buildings, roads. |
| 5 | Water-bodies | Rivers, canals, permanent open water, ponds, reservoirs. |
| 6 | Arable land | Land capable of being ploughed, pasture land, temporary fallow land. |
| 7 | Bare land | Exposed soils and barren areas influenced by human impact. |
| 8 | Sandy area | Land covered with sand, sea beaches. |
| 9 | Tidal mudflat | Coastal wetlands that form when the mud is deposited by tides. |

In order to improve the classification accuracy, testing of previews and repeated segmentation of individual land cover classes were conducted. The same band set—near infrared (0.845–0.885 μm), red (0.630–0.680 μm), and green (0.525–0.600 μm) (band 5, 4, 3)—was used for SVM classification [24,27,41,42] to reduce the bias introduced when combining different bands. The LULC conversion areas and their proportions for 2017, 2018, and 2019 classified images were consequently derived from the classification results via ENVI 5.3. The thematic change workflow method was executed to distinguish 2017–2018, 2018–2019, and 2017–2019 land cover alterations. In order to calculate the transformation of land cover alterations from one type to another, a transitional probability matrix was used.

### 3.4. Per Capital Greening Area (PCGA)

We created the PCGA to calculate the refugee population-wise vegetative cover in each of the thirty-four refugee camps in sequence. The equation used in this research to compute the PCGA is as follows:

$$\text{PCGA} = \frac{\text{Vegetative Cover (Ha)}}{\text{Refugee Population}} \tag{1}$$

Here, PCGA datasets are the ratio between each of the thirty-four refugee camps vegetative cover and the number of refugees in 2017, 2018, and 2019. The PCGA dataset was created to identify highly vulnerable vegetation cover refugee camps in three-time periods, one before the refugee crisis in 2017 and two after the crisis broke out in 2018 and 2019. In the next part, the procedure deploying k-means classification to identify refugee camps with highly vulnerable vegetative cover is discussed.

### 3.5. Vegetative Vulnerable Refugee Camp Identification Based on PCGA Dataset Using K-Means Classification

The k-means classification technique permits a deeper understanding of the vegetative cover capacity changes due to the sudden influx of refugees at each of the thirty-four refugee camps, and eventually identification of the refugee camps with the highly vulnerable vegetation cover over the study period. K-means classification identifies observations that are alike for categorization [29,43–45]. The analysis was run on R studio and associated packages; it is an open-source statistical computing and graphics software.

The distance measurement in clustering defines how the similarity of two non-spatial PCGA observations ($x_i$, $x_j$) is calculated and influences the shape of the cluster. In this

analysis, the Euclidean distance was computed to represent the dissimilarity between each pair of observations, and the equation is as below:

$$Dis(x_i, x_j) = \sqrt[2]{\left\{ \left(x_i^1 - x_j^1\right)^2 + \left(x_i^2 - x_j^2\right)^2 + \left(x_i^3 - x_j^3\right)^2 \right\}} \qquad (2)$$

where $x_i$ is the PCGA observation with three variables: $x_i^1$, $x_i^2$ and $x_i^3$, representing PCGA of the three study periods 2017, 2018, and 2019. There are thirty-four observations in total, where $(x_i, x_j)$ $(i = 1, 2, \ldots, 34; j = 1, 2, \ldots, 34)$ represents a pair of the yield observations, and $Dis(x_i, x_j) = 0$ whenever $i = j$.

Here, in the PCGA datasets, there are thirty-four observations with three variables. The elbow method, silhouette method, and gap statistics are the three popular methods for determining the optimal clusters [33]. The most popular is the elbow method and this was also used in this research. We plotted the curve of the total Within-cluster Sum of Square (WSS) according to the number of clusters k (in this case, k = 10). In general, the location of a bend (knee) in the plot is considered as an indicator of the appropriate number of clusters. In this research, we used the standard algorithm, namely the Hartigan–Wong algorithm, which holds several potential advantages compared to the classical optimization heuristic, Lloyd's algorithm [43].

## 4. Results

*4.1. Rapid Declining of Vegetative Cover and Increased Settlement and Bare Land in Ukhiya-Teknaf, the Situation of Pre and Post Rohingya Refugee Crisis, 2017–2019*

The confusion matrix method using ground truth Region of Interest (ROI) for each of the nine classes was applied by segregating test pixels to the corresponding location in the classified images [19,30,46]. The reference data (788 polygons and 8219 pixels for 3 Landsat OLI images) were manually selected to assess image classification accuracy. The producer and user accuracy of the three classified images were obtained from confusion matrix techniques. Overall classification accuracy for 2017, 2018, and 2019 is 99.51%, 98.16%, and 96.36%, with kappa coefficient index values of 0.98, 0.97, and 0.98, respectively (see Table 3).

**Table 3.** Image classification accuracy verification values of 2017, 2018, and 2019 (here, PA = Producer's accuracy and UA = User's accuracy).

| LULC Classes | 2017 | | 2018 | | 2019 | |
|---|---|---|---|---|---|---|
| | PA (%) | UA (%) | PA (%) | UA (%) | PA (%) | UA (%) |
| Vegetation | 98.92 | 97.58 | 98.99 | 98.99 | 100 | 99.39 |
| Settlement | 95.59 | 98.48 | 76.38 | 97.98 | 94.49 | 92.31 |
| Water-body | 99.8 | 99.8 | 100 | 100 | 100 | 99.57 |
| Agricultural land | 84.62 | 92.63 | 93.92 | 93.92 | 93.33 | 100 |
| Aquaculture | 98.46 | 99.22 | 99.1 | 100 | 95.83 | 96.99 |
| Arable land | 100 | 97.92 | 100 | 94.81 | 100 | 98.77 |
| Tidal mudflat | 99.11 | 99.11 | 99.69 | 100 | 96.54 | 96.88 |
| Sandy area | 100 | 99.3 | 99.74 | 95.98 | 97.75 | 98.09 |
| Bare land | 94.12 | 100 | 99.07 | 100 | 98.02 | 99.5 |
| Overall Accuracy | 98.51% | | 98.16% | | 96.36% | |
| Kappa Coefficient | 0.98 | | 0.97 | | 0.98 | |

The classification results of the classified images of Ukhiya-Teknaf in 2017, 2018, and 2019 are presented in Figure 5.

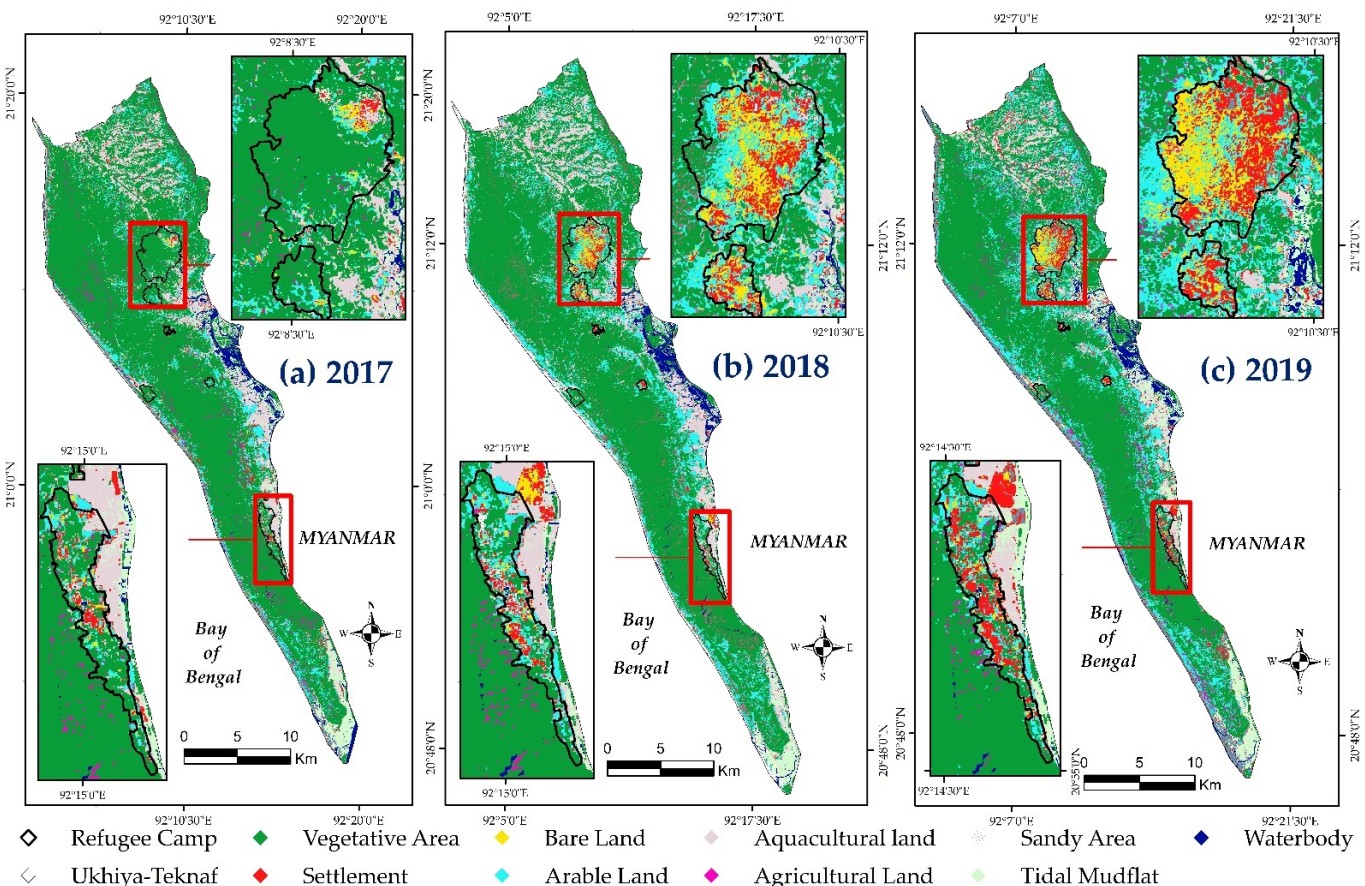

**Figure 5.** Land cover maps for the study area, Ukhiya-Teknaf region, classified into nine major land cover classes including: vegetation (green), bare land (yellow), agricultural land (light blue), sandy area (grey), water-body (dark blue), settlement (red), arable land (cyan), agricultural land (violet), and tidal mudflat, at three time-steps (**a**) 2017, (**b**) 2018, and (**c**) 2019.

The statistics of different types of land use areas and their proportions are presented in Figure 6. By visual interpretation, it is evident that the vegetative cover of the study area has tested a dynamic (transition and conversion) relationship after the refugee crisis broke out. The vegetative cover is the largest land cover area and declined from 67.87% to 58.37% over the study period.

Table 4 demonstrates the net land cover differences of each class in 2017–2018, 2018–2019, and 2017–2019. The most notable land cover changes were the rapid decrease of vegetative cover (3359 hectares/8.58%), an associated increase of agricultural land (419.7 hectares/34.4%), arable land (2670.8 hectares/55.88%), settlement (356.9 hectares/27.02%), and bare land (264.9 hectares/70.15%) in 2017–2018. Similarly, the rapid decrease of vegetative cover (2123.2 hectares/5.93%) and agricultural land (189 hectares/11.53%), an associated increase of arable land (2194.4 hectares/29.46%), settlement (1079.4 hectares/64.33%), and bare land (111 hectares/17.28%) in 2018–2019. Overall, between 2017 and 2019, a massive decrease of vegetative cover (5482.2 hectares/14%), and the associated increase of settlement (1436.3 hectares/108.74%), arable land (4865.2 hectares/101.8%), bare land (375.9 hectares/99.5%), and agricultural land (230.7 hectares/18.91%) was indicated.

Tables A3–A5 show the spatiotemporal land cover changes in 2017–2018, 2018–2019, and 2017–2019, respectively, from each land cover to another. Vegetative cover changes in the thirty-four Rohingya refugee camps were quantified based on three classes such as vegetative cover, settlement, and non-vegetative cover in 2017–2019, as discussed in the next section.

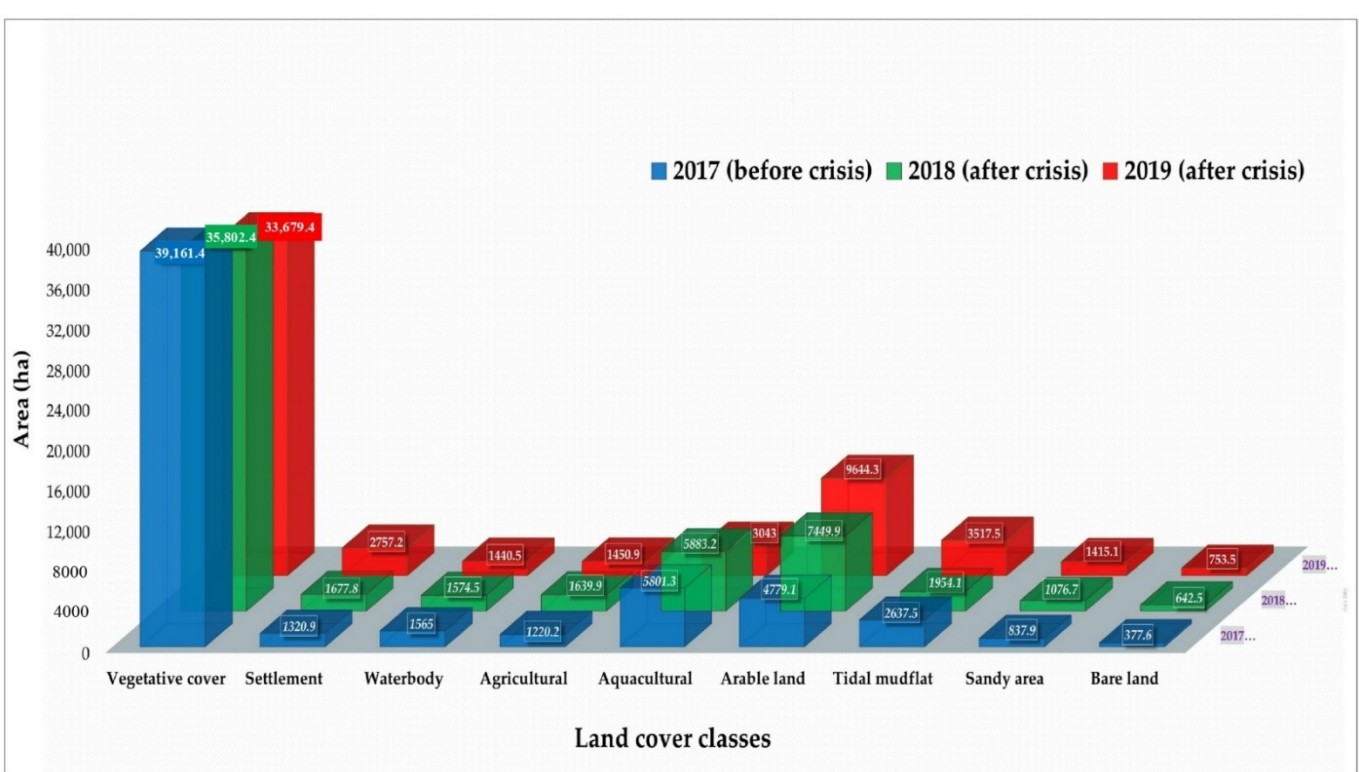

**Figure 6.** Land cover area (in thousands of hectares) changes in Ukhiya-Teknaf region before and after the Rohingya refugee crisis between 2017–2019, by the period of observation.

**Table 4.** Gain and loss of different LULC areas (in hectares and percentage) in Ukhiya-Teknaf, pre and post refugee crisis, by the period of observation.

| LULC Classes | 2017–2018 | | 2018–2019 | | 2017–2019 | |
|---|---|---|---|---|---|---|
| | **Area** | **%** | **Area** | **%** | **Area** | **%** |
| Vegetative | −3359 | −8.58 | −2123.2 | −5.93 | −5482.2 | −14 |
| Settlement | 356.9 | 27.02 | 1079.4 | 64.33 | 1436.3 | 108.74 |
| Waterbody | 9.5 | 0.61 | −134 | −8.51 | −124.5 | −7.96 |
| Agricultural Land | 419.7 | 34.4 | −189 | −11.53 | 230.7 | 18.91 |
| Aquaculture land | 81.9 | 1.41 | −2840.2 | −48.28 | −2758.3 | −47.55 |
| Arable land | 2670.8 | 55.88 | 2194.4 | 29.46 | 4865.2 | 101.8 |
| Tidal mudflat | −683.4 | −25.91 | 1563.4 | 80.01 | 880 | 33.36 |
| Sandy area | 238.8 | 28.5 | 338.4 | 31.43 | 577.2 | 68.89 |
| Bare land | 264.9 | 70.15 | 111 | 17.28 | 375.9 | 99.55 |

*4.2. Expansion of Rohingya Refugee Settlement and Decline of Vegetative Cover among All Thirty-Four Refugee Camp Areas, the Situation of Pre and Post Rohingya Refugee Crisis, 2017–2019*

The refugee settlement expanded at a massive rate across the thirty-four refugee camps, increasing from 101 hectares to 822 hectares between January 2017 and February 2019, with a total growth rate of 717% (see Table 5). In this analysis, the vegetative covers of the existing thirty-four refugee camps show a rapid downward trend, from 1866.33 hectares to 381.33 hectares, with a declining rate of 79.57%.

The conversion matrix of land cover in 2017–2019 suggests that vegetative to non-vegetative (i.e., waterbody, agricultural land, aquaculture land, arable land, tidal mudflat, and sandy area) land cover increased rapidly, accounting for 956 hectares; additionally, the vegetative to settlement and non-vegetative to settlement conversion area was 546 hectares and 209 hectares. The total net vegetative, non-vegetative, and settlement cover changes in all thirty-four camps are −1502.56 hectares, +760.89 hectares, and +729.99 hectares. In

January 2017, 74.36% of the refugee camp area was covered by vegetative land, indicating that vegetative was the dominant land cover before the crisis began. However, after the refugee crisis broke out, a massive decline of vegetative cover was recorded in March 2019, accounting for only 15.19%. The refugee settlement was the smallest land cover in 2017, accounting for only 4.01%, but it increased by 32.75% in 2019. The non-vegetative cover accounted for 21.64% in 2017 and was estimated to be 52.06%; it is the dominant land cover after the refugee crisis in 2019.

**Table 5.** Area and spatial changes in land cover classes and overall net gain and losses between 2017 and 2019 among the thirty-four Rohingya refugee camp areas.

| LULC Classes | 2017 (ha) | (%) | 2019 (ha) | (%) | Class Change (ha) | Growth/Decline Rate (%) | Net Change in Camp Area (ha) |
|---|---|---|---|---|---|---|---|
| Vegetative | 1866.33 | 74.36 | 381.33 | 15.19 | −1485 | −79.57 | −1502.6 |
| Settlement | 100.62 | 4.01 | 822.06 | 32.75 | 721 | 717 | 729.99 |
| Non-Vegetative | 543.06 | 21.64 | 1306.62 | 52.06 | 763 | 958.93 | 760.89 |

Appendix A, Table A1 shows the overall net changes of different land cover between 2017 and 2019, among thirty-four Rohingya refugee camps. The most significant net vegetative cover changes occurred in Camp 4, Camp 17, Camp 8W, Camp 18, and Camp 20 (ext.), accounting for 116.55 hectares, 96.75 hectares, 78.12 hectares, 74.52 hectares, and 75.78 hectares, respectively, between 2017 and 2019 (see Figure 7).

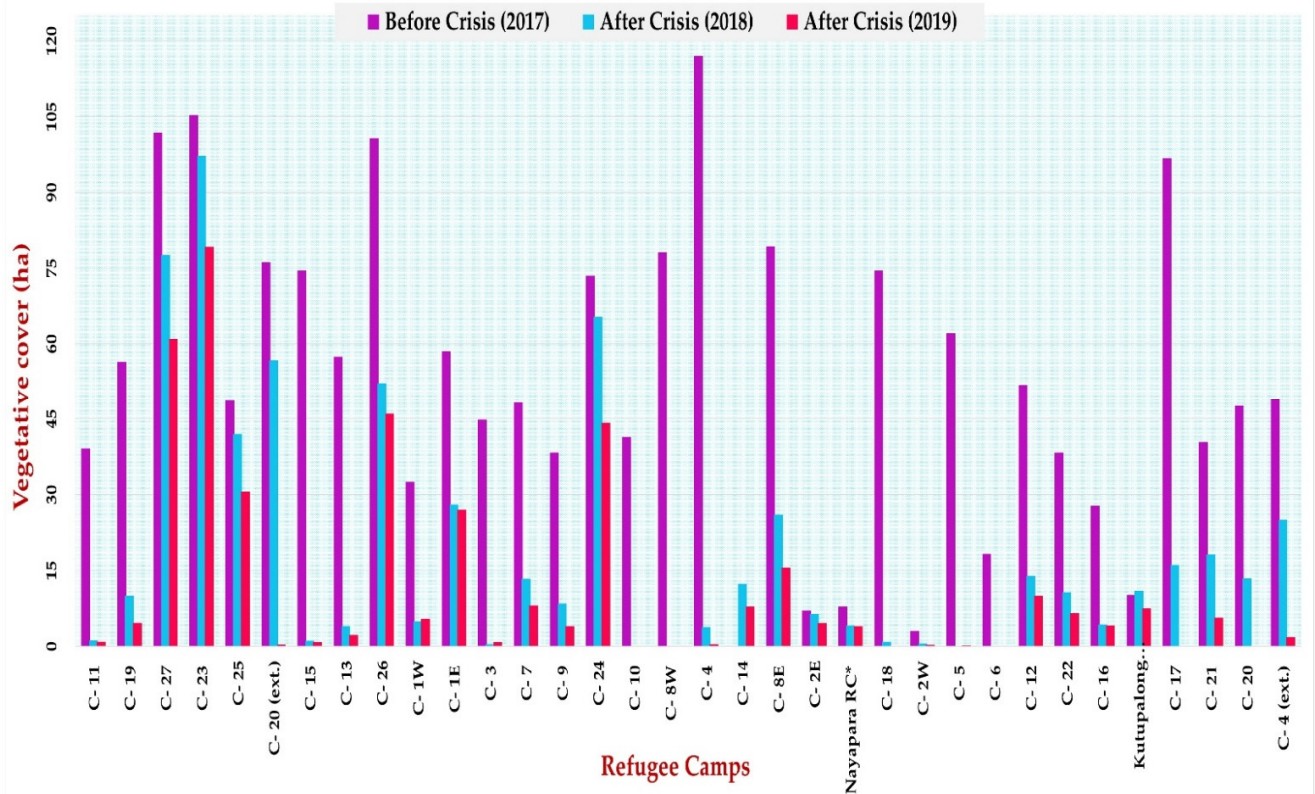

**Figure 7.** Camp-wise vegetative cover changes, pre and post refugee crisis (between 2017 and 2019), among thirty-four refugee camps in Ukhiya-Teknaf. Here, C = camp; RC* = registered camp; Ext. = extension.

The massive influx of refugee populations in a short period made the vegetative cover in the refugee camp extremely vulnerable. In order to understand the vegetative capacity and identify the refugee camps with highly vulnerable vegetation cover, in this study, we used the PCGA dataset and k-means classification.

### 4.3. Vegetative Cover Vulnerable Rohingya Refugee Camp Identification Using K-Means Classification

The PCGA dataset of each of the thirty-four refugee camps in 2017, 2018, and 2019 is plotted in Figure 8, where the differences in PCGA before and after the Rohingya refugee crisis are massive. However, to understand the changing pattern of PCGA from a qualitative perspective as well as identify refugee camps with highly vegetative cover that is vulnerable, we used k-means classification, aiming to divide the PCGA observation data into cluster groups.

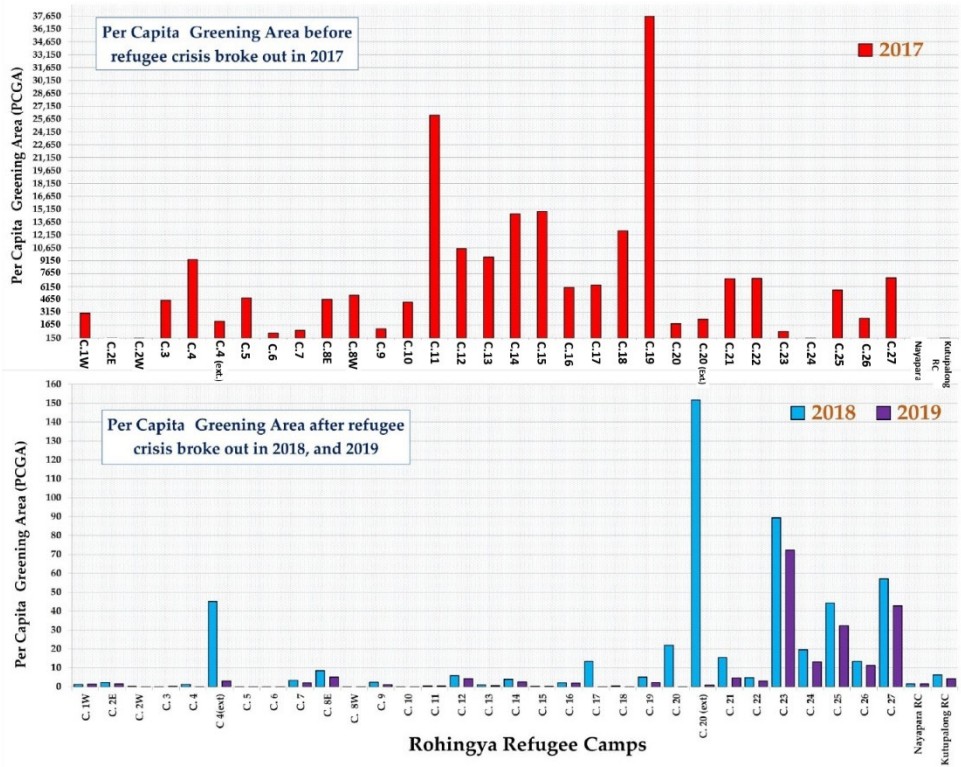

**Figure 8.** Per capital greening area (in hectares) of thirty-four Rohingya refugee camps in Ukhiya-Teknaf of Bangladesh, pre and post refugee influx, between 2017 and 2019.

The normalized dissimilarity result of each observation is demonstrated in Figure 9 and the data presented in Appendix A, Table A2. The higher values indicate a more substantial dissimilarity than lower.

The optimal cluster number was determined using the elbow method, and in this case, the optimal cluster number is three, as is shown in Figure 10. In K-means clustering, each cluster is represented by its center (i.e., centroid), which corresponds to the mean of the points assigned to the cluster [31].

Since the number of the cluster must be set before running the algorithm, it is often advantageous to use several different values of k and examine the differences in the result [45]. However, in this analysis, different values of k (2, 3, 4, 5, and 6) were used before setting the optimal cluster number as three—the clustering result and clustering vector group presented in Figure 11 and Table 6.

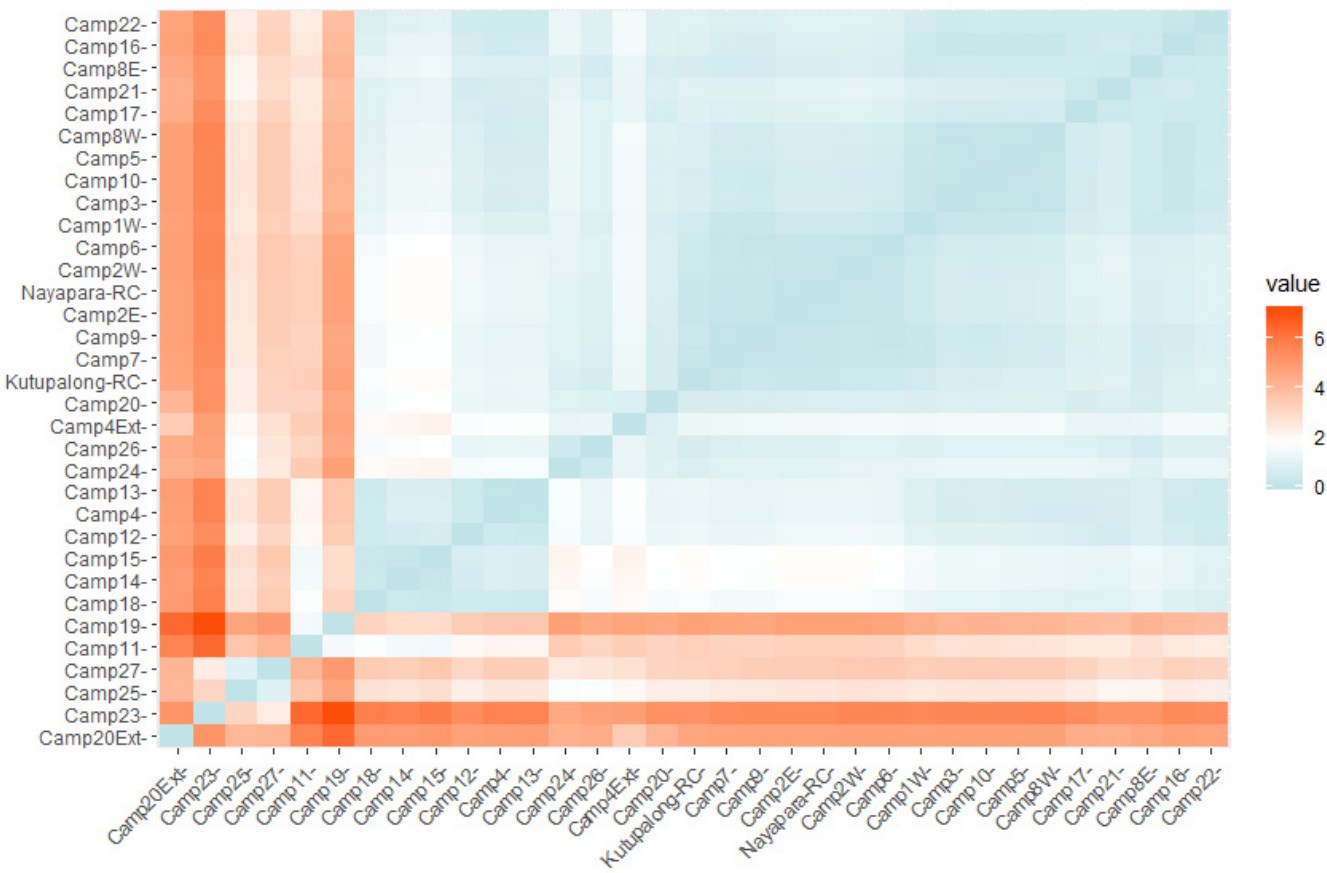

**Figure 9.** Heat map of PCGA dissimilarity, by all thirty-four Rohingya Refugee Camps, from 2017–2019. A higher value represents significant dissimilarity, and the lower value represents a dataset that appears to be reasonably similar.

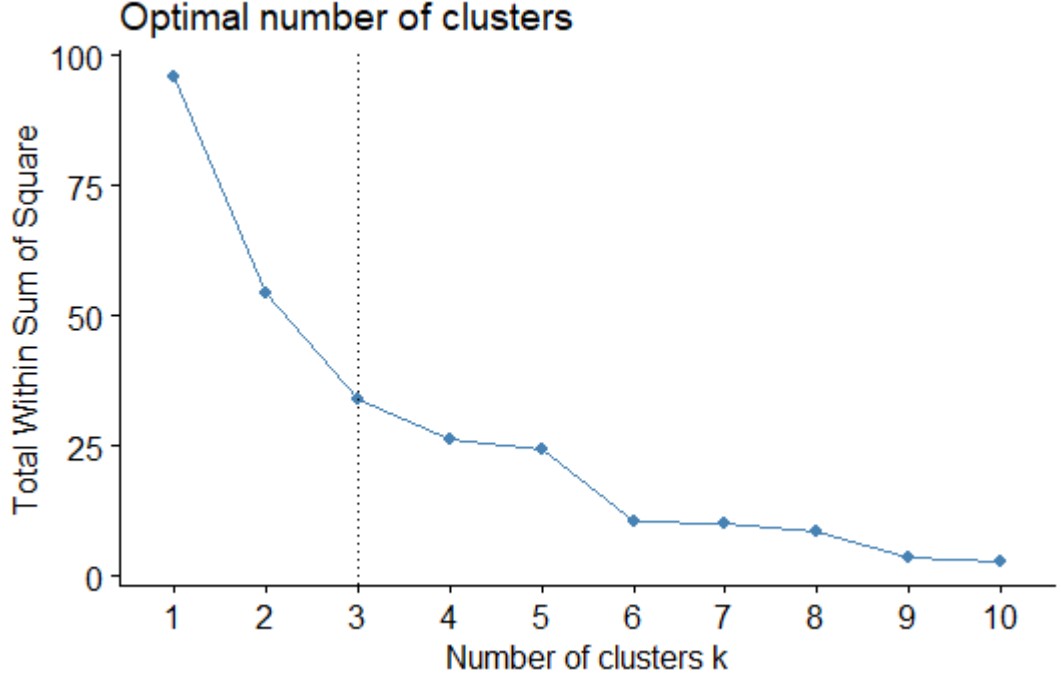

**Figure 10.** Determination of the optimal number of the cluster for vegetative cover vulnerability analysis (the optimal cluster number is 3, as it appears to be the bend in the knee/elbow).

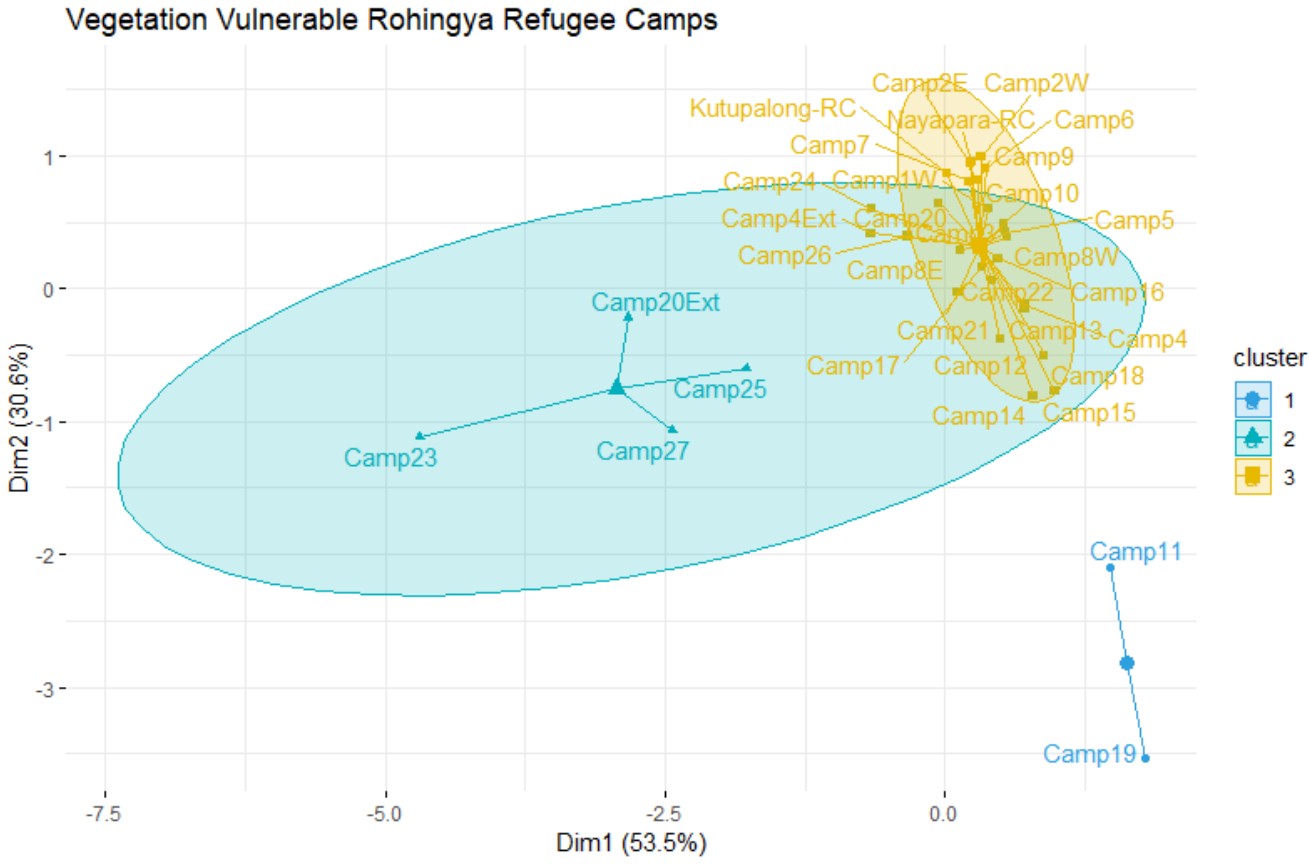

**Figure 11.** Clustering results of vegetative vulnerable refugee camps in Ukhiya-Teknaf in Bangladesh between 2017 and 2019. The results show the PCGA dataset grouped into three clusters. Besides, dim1 (53.5%) and dim2 (30.6%) explain that 53.5% variance within the PCGA dataset is captured by Principal Component Analysis 1 (PCA 1), while PCA 2 demonstrates 30.6% variance of the dataset.

**Table 6.** K-means classification results of PCGA, clustering vector groups, mean values, and the average mean value of each cluster, lower values represent high vulnerability, and higher values represent low vulnerability.

| Cluster | 2017 (Mean) | 2018 (Mean) | 2019 (Mean) | Average (Mean) | Clustering Vector |
|---|---|---|---|---|---|
| 2 | −0.31 | 2.22 | 2.05 | 1.32 | Camp23, Camp27, Camp25, Camp20 (extension) |
| 1 | 3.21 | −0.42 | −0.34 | 0.82 | Camp19, Camp11 |
| 3 | −0.19 | −0.30 | −0.28 | −0.26 | Camp 1E, Camp 1W, Camp 2E, Camp 2W, Camp 3, Camp 4, Camp 4 (extension), Camp 5, Camp 6, Camp 7, Camp 8E, Camp 8W, Camp 9, Camp 10, Camp 12, Camp 13, Camp 14, Camp 15, Camp 16, Camp 17, Camp 18, Camp 20, Camp 21, Camp 22, Camp 24, Camp 26, Nayapara-RC, and Kutupalong-RC |

Figure 11 and Table 6 demonstrate the vegetative cover vulnerability of refugee camps over the study period. The result indicates that Camp 20 (extension), Camp 23, Camp 25, and Camp 27 belong to cluster 2, and the average mean is 3.95, suggesting less vulnerable

refugee camps. Similarly, Camp 11 and Camp 19 are included in cluster 1, and the average mean is 2.45 which states that camp vulnerability is moderate. Likewise, Camp 1W, Camp 2E, Camp 2W, Camp 3, Camp 4, Camp 4 (extension), Camp 5, Camp 6, Camp 7, Camp 8E, Camp 8W, Camp 9, Camp 10, Camp 12, Camp 13, Camp 14, Camp 15, Camp 16, Camp 17, Camp 18, Camp 20, Camp 21, Camp 22, Camp 24, Camp 26, Nayapara-RC, and Kutupalong-RC are linked with cluster 3. The refugee camps under cluster 3 are highly vegetative cover vulnerable, and the average mean is −0.77. This study affirms that the rapid expansion of the Rohingya refugee settlement makes about 82% of Rohingya refugee camps' vegetative covers highly vulnerable.

## 5. Discussion

### 5.1. Land with Vegetative Cover Is the Primary Source of the Newly Increased Settlement and Bare Areas

To accommodate the mass influx of Rohingya refugees, approximately 5482.2 hectares of vegetative cover in and surrounding the refugee camp was razed for settling thirty-two new non-registered refugee camps. The mass decline in vegetative cover was found in and surrounding the greatest concentration of Rohingya refugee settlements [2]. Furthermore, the large-scale deforestation mainly took place towards the south-west direction of pre-existing Kutupalong RC and the north-south direction of Nayapara RC (see Figure 12). Such an unprecedented mass decline of vegetative cover puts severe pressure on the socio-economic fabric as well as the ecology not limited to Ukhiya-Teknaf but the entire country [47].

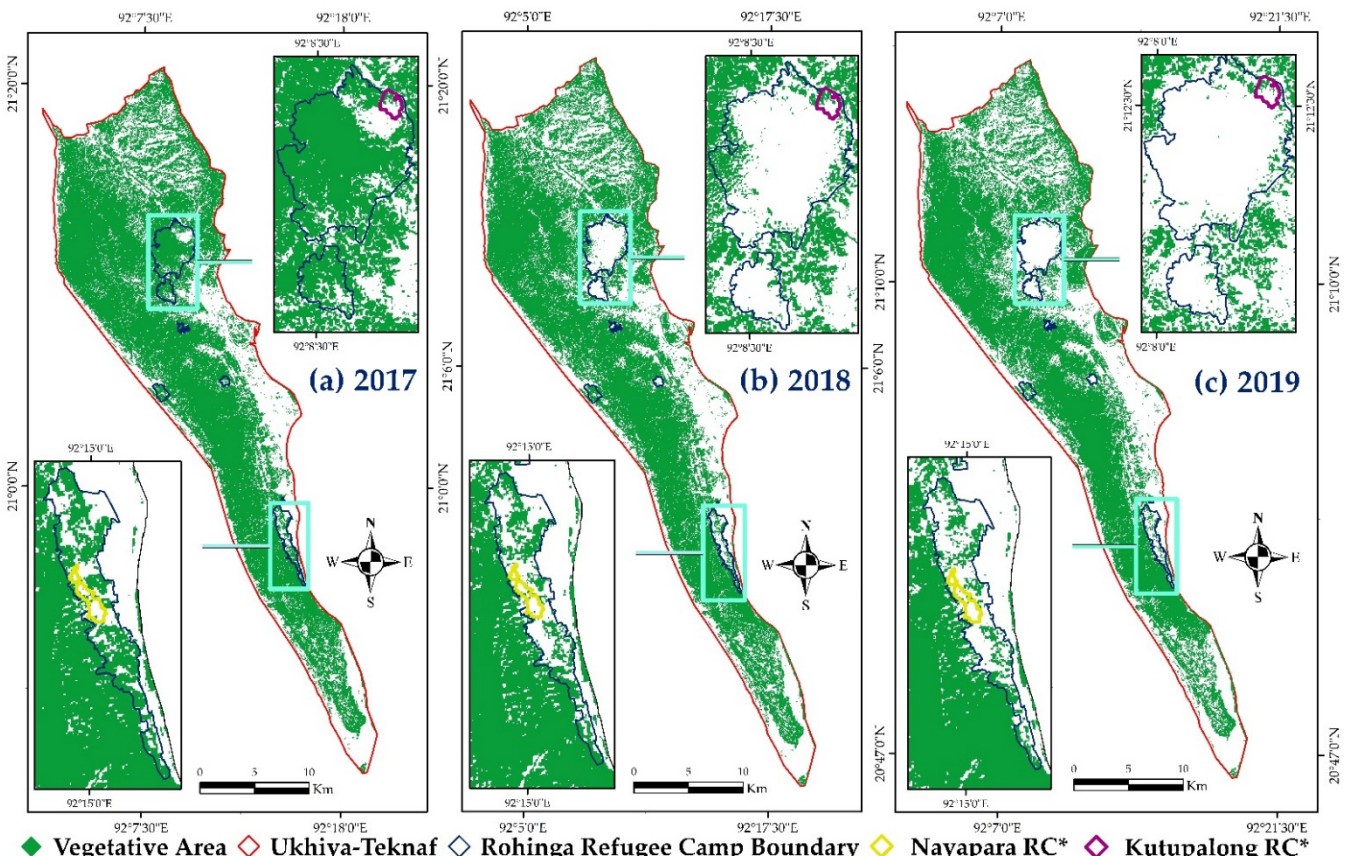

**Figure 12.** Changes of vegetative cover, Ukhiya-Teknaf, 2017–2019. (**a**) 2017; (**b**) 2018; and (**c**) 2019 (Out of 34 refugee camps there are only two registered refugee camps, which is indicated by the symbol *).

The host community, national and international NGO'S, as well as previously settled Rohingya refugees, had wiped out vegetative cover in Ukhiya-Teknaf for temporary

makeshift homes for the refugees, resulting in a rapid increase of land with settlement, arable and bare areas [2,4,9] (see Figure 13).

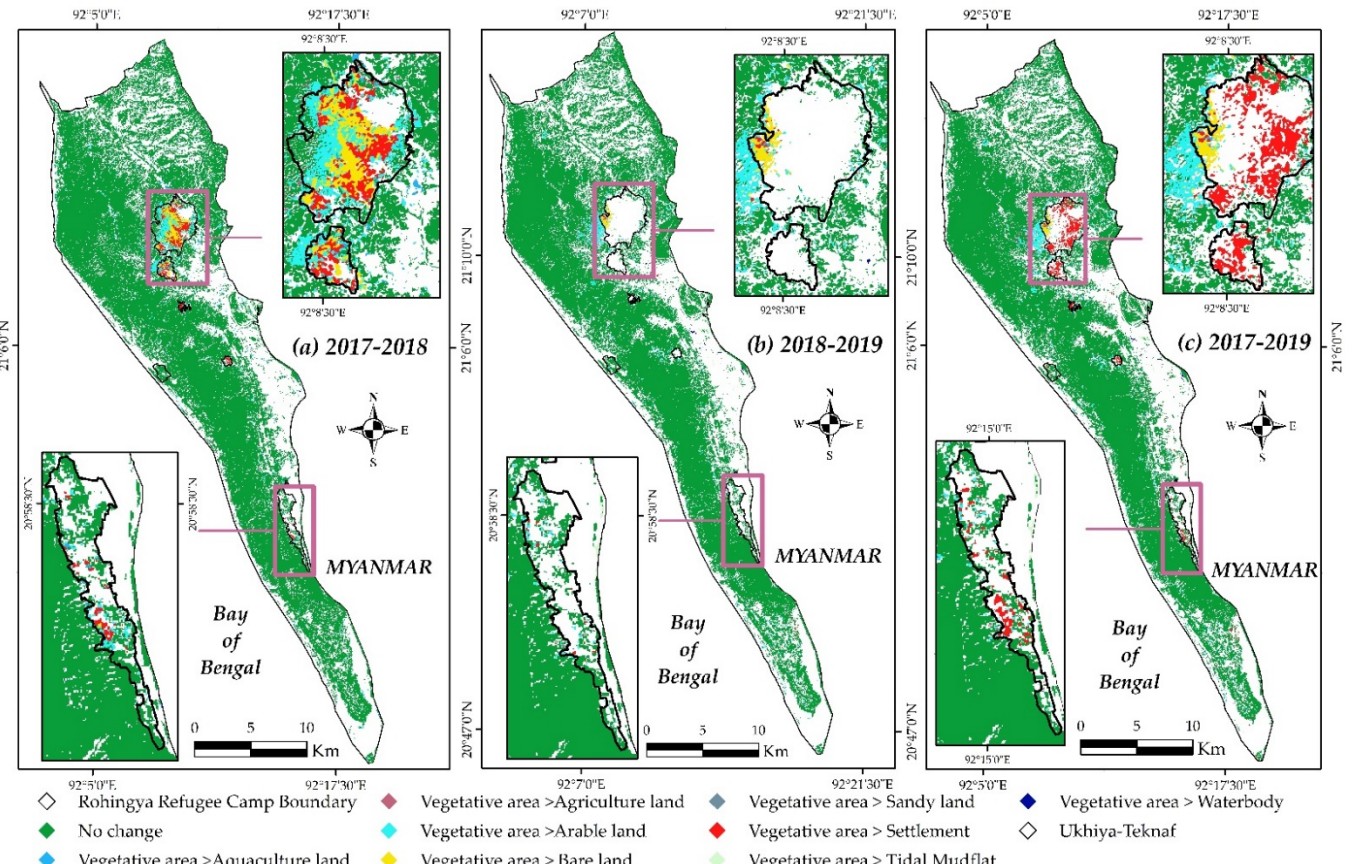

**Figure 13.** The transition of vegetation covers to other land use/land cover, Ukhiya-Teknaf. (**a**) 2017–2018, (**b**) 2018–2019, and (**c**) 2017–2019.

On the contrary, there was no significant conversion from other land covers to vegetative cover seen in and surrounding the refugee camps since the crisis broke out in 2017 [5,38], see Figure 14.

Bangladesh is a tiny country by area, and already struggling to solve poverty and overpopulation problems of its own, associated with ever-increasing environmental and climatic risks [48]. The government has no luxury of open land to construct a new settlement for nearly millions of refugees and accommodate them therein [12]. The incoming refugee's magnitude and rate have already created enormous pressure on the natural resources and ecological environment around the refugee camps and substantially alerted the local landscape. The massive vegetative cover loss before and after the refugee crisis is identified by red color in Figure 15.

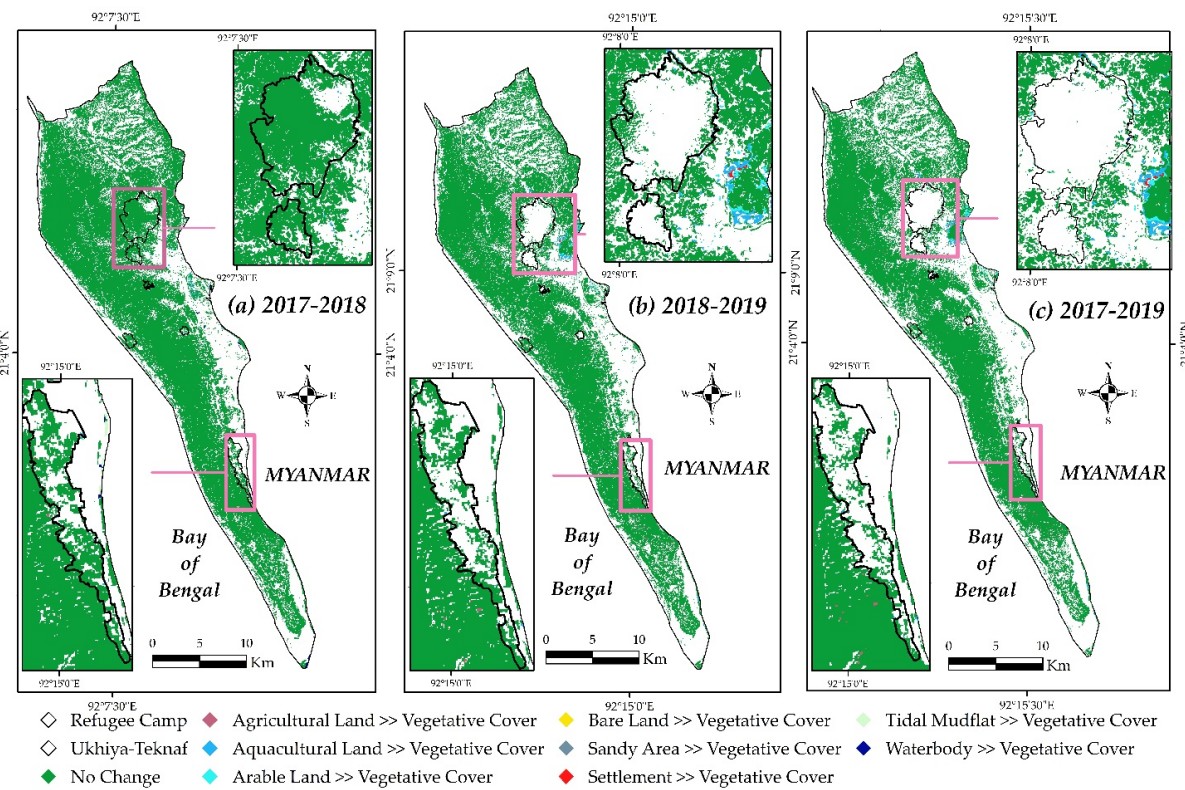

**Figure 14.** Conversion of vegetative cover from other land use/land cover, Ukhiya-Teknaf. (**a**) 2017–2018, (**b**) 2018–2019, and (**c**) 2017–2019.

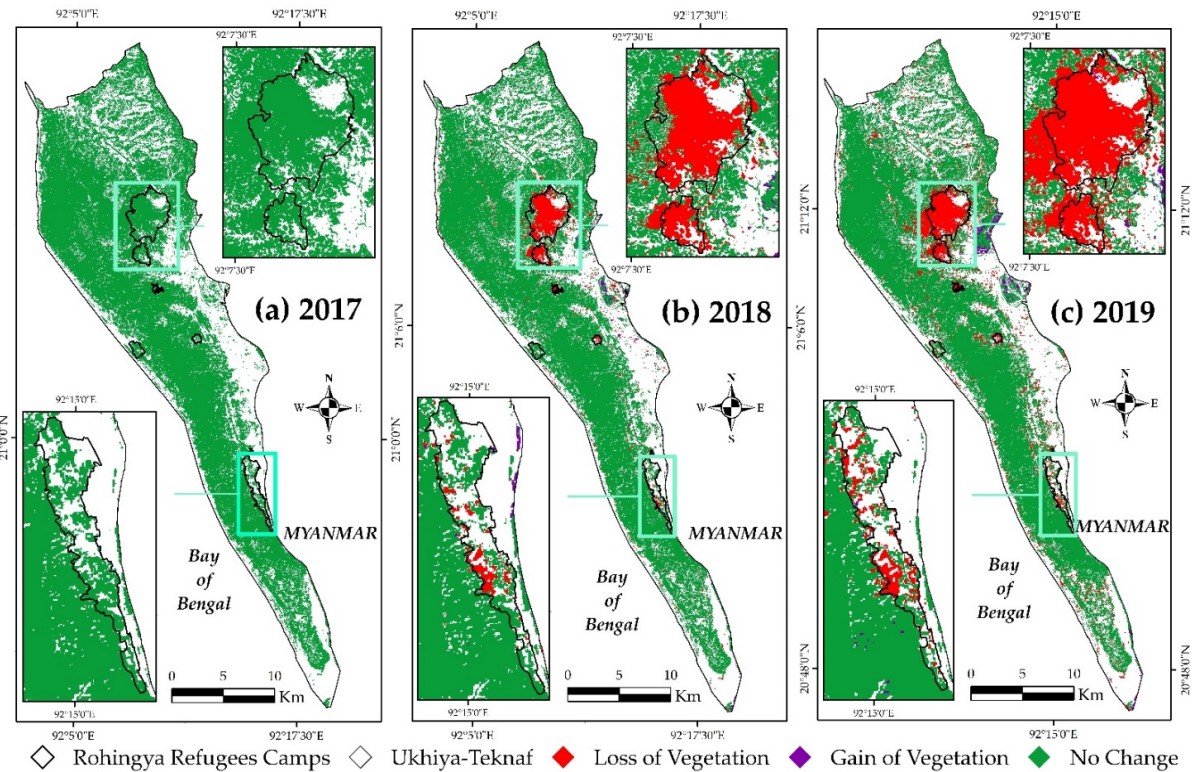

**Figure 15.** Spatiotemporal changes of vegetative cover, pre and post refugee crisis, Ukhiya-Teknaf: (**a**) Vegetative cover before the refugee influx in 2017, (**b**) loss of vegetative cover after the refugee influx in 2018, (**c**) rapid loss of vegetative cover due to the refugee influx in 2019. Green, red, and violet colors represent no change, total loss, and total gain, respectively, by the period of observation.

Besides the rapid influx of Rohingyas, different activities such as clearing land for building new camps [15], expansion of agricultural land [4], cutting down the forest for fuelwood collection [2], and so on are the critical factors behind the decline of vegetative cover in Ukhiya-Teknaf. The refugee camps demand 750,000 kg of fuelwood each day, and to meet this substantial demand, they razed down in and surrounding protected forest [2]. In addition, nearly a million refugees have accommodated a total area of only 2510.01 hectares of hilly land. As a consequence, rampant hill wiping out the herbaceous layer may trigger landslides during the rainy season. Landslides are a frequent natural disaster in the hilly mountain areas in Bangladesh and cost many lives each year [14]. A gigantic landslide in the campsite might prompt a more significant humanitarian situation. Though the government of Bangladesh strictly prohibited the expansion of the refugee camp further to protect the reserve forest, many refugees are camping in the deep forest and blocking the elephant corridors [38]. Severe deforestation is resulting at local and regional levels to meet the demand within the refugee camps. The international organization for migration estimated 3000 out of 43,000 acres of forest land in the hilly district of Cox's Bazar has been destroyed by the refugees since the crisis broke out, resulting in a rapid increase of refugee settlements and bare areas [9].

### 5.2. Nearly 82% of Rohingya Refugee Camps Land with Vegetative Covers Are Highly Vulnerable

It is evident that the mass refugee population has caused rapid and long-term negative impacts on the environment in and surrounding the refugee camps of host region Ukhiya-Teknaf, Bangladesh. To accommodate these millions of refugees, with the help of local and international volunteer organizations, the Bangladesh government allowed thirty-four big and small refugee camps, varying from 49,468 (in Camp 15) to 4630 refugees (in Camp 20-extension), to be built [35]. As a result, approximately 1502.56 hectares of vegetative cover was lost among the thirty-four camps between 2017 and 2019 (see Figure 16).

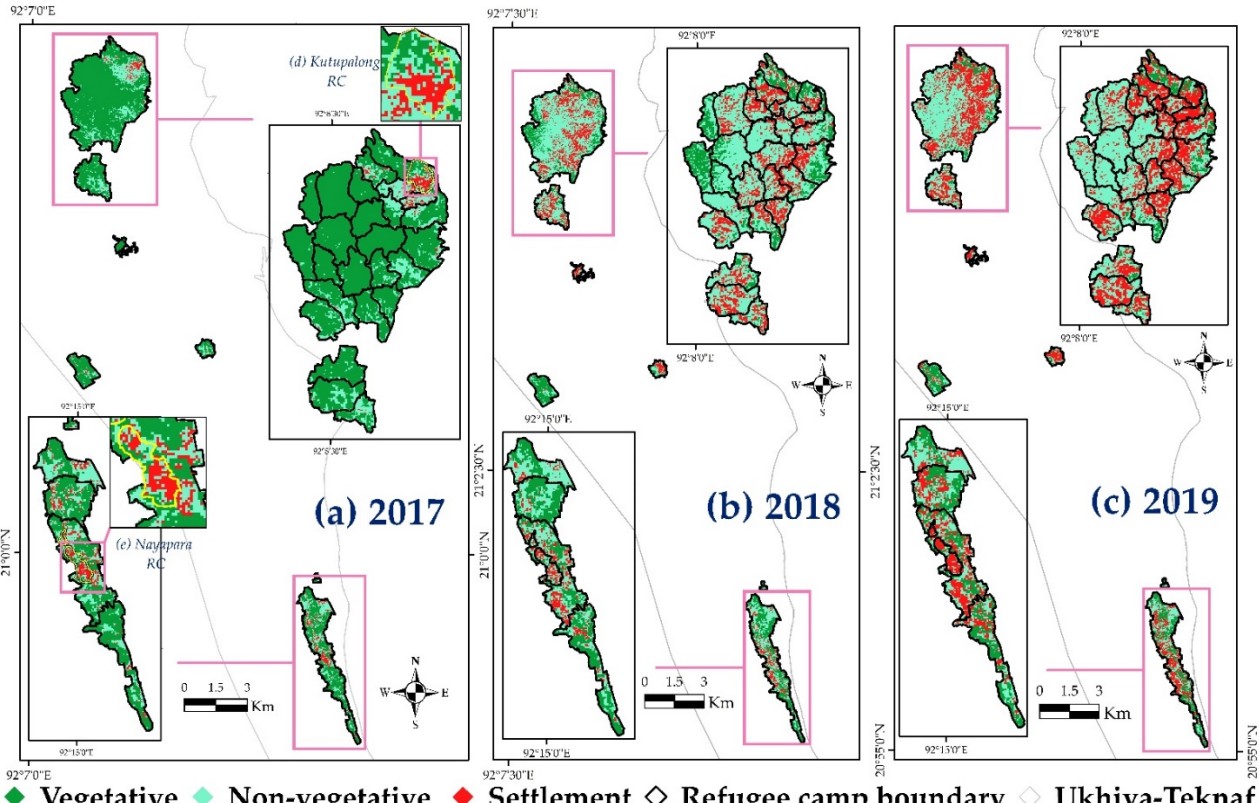

**Figure 16.** Land cover maps of the thirty-four Rohingya refugee camp areas classified into three major land cover classes, including vegetative, non-vegetative, and settlement represented as green, cyan, and red, respectively, at three time-periods, representing pre-influx (**a**) January 2017, and post-influx (**b**) March 2018, and (**c**) February 2019.

The thirty-four Rohingya refugee camps were settled near a highly sensitive ecological region, containing a protected forest for endangered wild animals. The rapid spatial expansion of refugee settlements and corresponding human-made activities (such as cutting down the forest cover for fuelwood, timber, and other substance needs) poses a severe threat to wildlife as well as the surrounding ecosystem. This vast expansion of refugee settlements and the decline of large-scale vegetative cover mainly took place on two neighboring pre-existing refugee camps, namely Kutupalong RC and Nayapara RC. Almost every day, new refugees are joining the refugee camp and threatening further sociological and environmental degradation. The ultimate goal of this research was to identify refugee camps with highly vulnerable vegetation cover. This study estimated 28 out of 34 (82.35%) Rohingya refugee camps with highly vulnerable vegetation cover, and most of them were expanded through south, west, and south-west direction from Kutupalong RC. If this trend continues, we fare to speak that, soon, there will be no vegetative cover in and surrounding the refugee camps.

The tension between the host community and refugees is increasing due to cultural differences and lack of labor opportunities, as well as an invasion of forest resources. Approximately 1.2 million people in the host population have been negatively affected due to the refugee crisis, and they are yet to receive minimum attention or support from the local government or international community [13]. The frequency of labor opportunities and wages has reduced since the crisis broke out. Approximately 1500 local inhabitants previously involved in the community forestry program face a loss of income [11]. The local community near refugee camps lost vegetable plots and agricultural fields due to set-up makeshift homes for refugees in the early stage. Some refugees, directly and indirectly, are involved in drug smuggling (especially "Yaba", imported from Myanmar), human and sex trafficking, prostitution, and robberies [2,4,15,38]. Over one million refugees changed the population configuration of Cox's Bazar district, one of the famous destinations among the local and international tourists and put a severe threat to the tourism sector of this region.

*5.3. The Bangladesh Government Might Relocate the Rohingya Refugees to the Sittwe and Take Initiatives to Establish the Whole Refugee Settlement Area as an "Ecological Park" Justifying Proper Guidelines and Protocols/Land Use Policy Recommendations*

The government of Bangladesh is spending 0.3 billion dollars to resettle nearly 100,000 Rohingya refugees at Bhashan Char to solve the dreadful overcrowding in the Rohingya refugee camps in Cox's Bazar [49], where about one million Rohingya refugees live. The Bangladesh government has already initiated the resettlement process and moved nearly 20,000 Rohingya refugees to Bhashan Char since December 2020. It is high time for the government and responsible bodies to prepare effective guidelines and policy, and to take initiatives, accordingly, with an aim to restore huge deforested land-mass of Ukhiya-Teknaf into nature.

The rapid exodus of Rohingya refugees in Ukhiya-Teknaf of Bangladesh since August 2017 triggered severe deforestation [2,14,15]. This requires a permanent solution to protect the biodiversity and ecology of the region. For instance, the government may take the initiatives of a peaceful return of the refugees to their origin Rakhine State, Myanmar. The government might give this strategic policy priority because a large sum of the Rohingya refugees want to go back to their mother state, Rakhine, but the guarantee of their safety and quality of life must be ensured. In that case, the Government of Bangladesh, along with its regional and international allies and organizations (such as UNHCR, WORLD BANK, SAARC, and UN), may put continuous pressure on the Government of Myanmar to ensure honor, safety, and the peaceful return of the Rohingya people to Rakhine state, Myanmar. The capital of Rakhine state, namely Sittwe, home of the Rohingya people, and the Teknaf Upazila, Cox's Bazar is a frontier state [12], and the distance between the two neighboring places is barely 101.56 km (see Figure 17). The government has spent 0.3 billion to resettle 100,000 people in Bhashan Char, and considering this as a reference class of cost estimation for the government, an estimated 3 billion US$ is needed to allocate to relocate one million Rohingyas to Rakhine State, Myanmar. Since it is a matter of time to relocate nearly one

million people to the Sittwe, the government may consider other areas similar to Bhashan Char as an alternative for temporary resettlement.

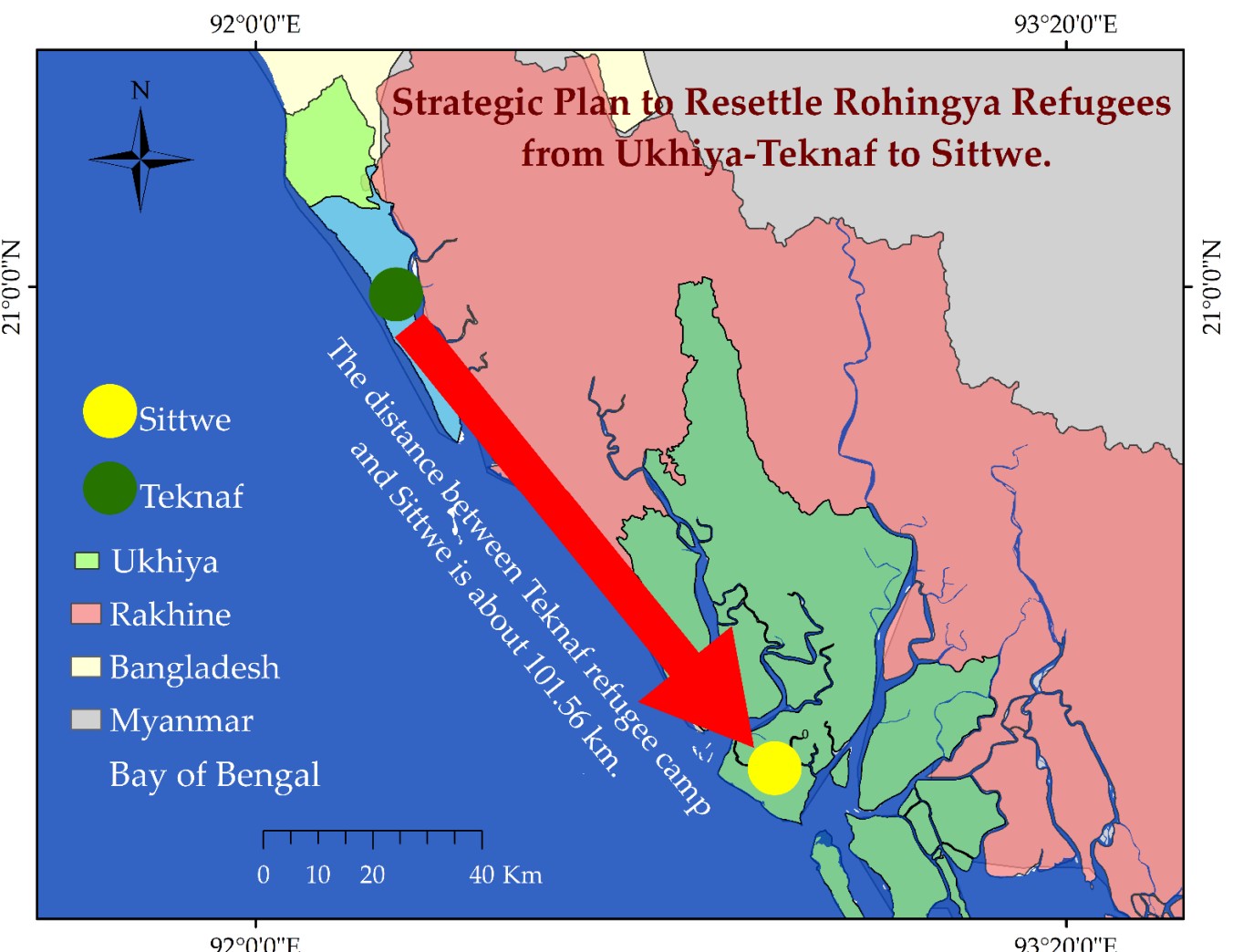

**Figure 17.** Strategic proposal to relocate Rohingyas from the densely populated refugee camps of Ukhiya and Teknaf to the capital of Rakhine state, Sittwe.

The sudden influx of millions of Rohingya refugees into the Ukhiya-Teknaf area has not only caused deforestation and damaged the ecological biodiversity in the refugee camp area but has put the entire region at risk [2]. The forest of Ukhiya-Teknaf is the habitat of many endangered flora and fauna as well as a habitat and breeding ground for various animals and birds; many of them are threatened with extinction [50]. In order to protect endangered flora, fauna, animals, and birds from extinction and maintain the ecological balance of the forest, the government and the responsible bodies might take the initiatives to establish the whole region as a national ecological park. The government may plan to set up separate sanctuaries for endangered flora, fauna, and birds in the region, bringing the region's biodiversity back to its former glory as well as being a great tourist destination for local and foreign visitors. In this regard, the government may take inspiration from the Alcatraz National Park, located 1.25 miles offshore from San Francisco, California, the United States, which was turned from a prison to a tourist attraction, a sanctuary for various flora and fauna, a recreation yard, art exhibition center, and so on [51]. The government of Bangladesh built the country's first eco-park in 2001 at an estimated cost of 1.2 million US$ on an area of 808 hectares located in Sitakunda Upazila, Chittagong [52]. The Ukhiya-Teknaf has 34 refugee camps, more than four times the size of the Sitakunda

eco-park (2510 hectares). Taking into account the estimated cost of the Sitakunda eco-park and the camp area as a reference class, the estimated cost of establishing an ecological park could be around 7.4–10 million US$.

The Chittagong hill track covers the largest area of forest in Bangladesh and the heart of this vast region is Ukhiya-Teknaf [53]. If the above-mentioned plan is implemented, it is possible to alleviate the damage done to the forest, biodiversity, and ecosystem after the arrival of Rohingyas in the area. The ultimate vision of this strategic policy is to protect the ecological biodiversity of the forest.

## 6. Conclusions

Since 25 August 2017, nearly a million Rohingya refugees fled into Bangladesh and put the environmentally fragile Ukhiya-Teknaf on the edge of massive ecological catastrophe. Based on supervised image classification techniques (such as SVM), and remote sensing data, this research quantified the temporal changes of vegetative cover between 2017 and 2019 (more specifically pre and post influx of Rohingya refugees) in Ukhiya-Teknaf and thirty-four refugee camps, respectively. We further identified the refugee camps with highly vulnerable vegetation cover utilizing the k-means clustering method and PCGA datasets. The results revealed that 79.57% vegetative cover was lost among the thirty-four refugee camps, and 14% vegetative cover was lost in the entire Ukhiya-Teknaf area since the refugee crisis broke out. Further, we found that 28 out of 34 refugee camps' vegetative covers are highly vulnerable. The vegetative cover is the primary source of the makeshift refugee camps in Ukhiya-Teknaf, and the unplanned and overcrowded refugee settlement seems to be one of the critical factors for the decline in vegetative cover around the thirty-four refugee camps. We show that there is an urgent need to estimate refugee camp settlements more precisely. The current use of Landsat 8 OLI seems to be insufficient to portray the evolution of the settlement in a desirable accuracy. VHR (Very High Resolution) satellite imagery would be a better choice in future research. Besides, the land cover of Ukhiya-Teknaf is dominated by meter-scale heterogeneity which is unlikely to be captured precisely by discrete classification methods. It might be interesting to consider subpixel spatial mixing models as a useful avenue of future work. By considering the ever-declining rate of vegetative cover, it will be too late to protect the fragile hilly forest, vegetative cover, and several rare endangered animals in Ukhiya-Teknaf if no initiatives are taken now or in the near future. Furthermore, the results of this research could be useful to the policymakers, planners, and researchers who are interested in utilizing these solutions for different studies.

**Author Contributions:** M.F.K. conceptualized the overall ideas of this study. M.F.K. and X.Z. designed the research. M.F.K. performed the data analysis and wrote the manuscript. M.F.K. and X.Z. were involved in improving the quality of this manuscript. X.Z. organized the funding, was the project administrator, and supervised the whole research. All authors have read and agreed to the published version of the manuscript.

**Funding:** This work was supported by the National Natural Science Foundation of China (Grant No. 41671384,41301410 and 41871377).

**Institutional Review Board Statement:** Not applicable.

**Informed Consent Statement:** Not applicable.

**Acknowledgments:** The authors are grateful to Orhan Altan, Stephen C McClure, Zhenfeng Shao for their continuous suggestions and support to improve this manuscript.

**Conflicts of Interest:** The authors declare no conflict of interest.

# Appendix A

**Table A1.** Vegetative cover changes and overall net changes of different LULC area (hectares) in 2017, 2018, and 2019, among thirty-four Rohingya refugee camps, by the period of observation. Here, V = Vegetative, R.C = Refugee Camp, N.V = Non-Vegetative, NVCC = Net Vegetative Cover Change, NRCC = Net Refugee Camp Change, NNVCC = Net Non-Vegetative Cover Change, NCARC = Net Change in All Refugee Camp Area, and C = Camp.

| Camp No | 2017(ha) | | | 2019(ha) | | | Net Change by Class (ha) | | | Net Change in All Refugee Camp Area (NCARC) |
|---|---|---|---|---|---|---|---|---|---|---|
| | V | R.C | N.V | V | R.C | N.V | NVCC | NRCC | NNVCC | |
| C. 1W | 32.49 | 4.23 | 17.64 | 5.49 | 33.93 | 14.94 | −27 | 29.7 | −2.7 | |
| C. 1E | 58.5 | 0.27 | 7.22 | 27 | 20.7 | 17.28 | −31.5 | 20.43 | 10.06 | |
| C. 2W | 3.06 | 5.31 | 31.95 | 0.27 | 22.5 | 17.55 | −2.79 | 17.19 | −14.4 | |
| C. 2E | 7.11 | 6.3 | 26.55 | 4.59 | 27 | 8.37 | −2.52 | 20.7 | −18.18 | |
| C. 3 | 44.82 | 0.36 | 1.17 | 0.8 | 21.6 | 23.94 | −44.02 | 21.24 | 22.77 | |
| C. 4 | 117 | 0.18 | 0.18 | 0.45 | 29.43 | 87.48 | −116.55 | 29.25 | 87.3 | |
| C. 4 (Ext.) | 48.87 | 0 | 0.72 | 1.8 | 5.4 | 42.39 | −47.07 | 5.4 | 41.67 | |
| C. 5 | 62.1 | 0 | 0.63 | 0.18 | 9.54 | 53.01 | −61.92 | 9.54 | 52.38 | |
| C. 6 | 18.27 | 0.18 | 18.81 | 0 | 22.5 | 14.76 | −18.27 | 22.32 | −4.05 | |
| C. 7 | 48.24 | 1.98 | 22.23 | 8.1 | 34.92 | 29.43 | −40.14 | 32.94 | 7.2 | |
| C. 8W | 78.12 | 0.09 | 0.09 | 0 | 33.21 | 45.03 | −78.12 | 33.12 | 44.94 | |
| C. 8E | 79.29 | 1.44 | 16.38 | 15.57 | 37.08 | 43.86 | −63.72 | 35.64 | 27.48 | |
| C. 9 | 38.34 | 0.09 | 27.27 | 3.96 | 34.29 | 27.46 | −34.38 | 34.2 | 0.19 | |
| C. 10 | 41.4 | 0.18 | 8.64 | 0 | 24.75 | 25.47 | −41.40 | 24.57 | 16.83 | |
| C. 11 | 39.15 | 0.09 | 8.91 | 0.9 | 27.09 | 20.16 | −38.25 | 27 | 11.25 | Camp Area |
| C. 12 | 51.75 | 0 | 12.06 | 9.99 | 17.37 | 36.45 | −41.76 | 17.37 | 24.39 | +729.99 ha |
| C. 13 | 57.42 | 0.9 | 17.55 | 2.25 | 35.55 | 38.07 | −55.17 | 34.65 | 20.52 | Vegetative Cover |
| C. 14 | 80.28 | 0 | 6.93 | 7.92 | 25.11 | 44.18 | −72.36 | 25.11 | 37.25 | −1502.56 ha |
| C. 15 | 74.52 | 0.27 | 24.39 | 0.81 | 49.77 | 48.6 | −73.71 | 49.5 | 24.21 | Non-Vegetative |
| C. 16 | 27.72 | 0.81 | 25.2 | 4.14 | 18.81 | 30.78 | −23.58 | 18 | 5.58 | +760.89 ha |
| C. 17 | 96.75 | 0 | 1.35 | 0.0001 | 7.02 | 91.08 | −96.75 | 7.02 | 89.73 | |
| C. 18 | 74.52 | 0 | 1.8 | 0 | 22.05 | 54.27 | −74.52 | 22.05 | 52.47 | |
| C. 19 | 56.43 | 0.27 | 21.06 | 4.68 | 12.69 | 60.39 | −51.75 | 12.42 | 39.33 | |
| C. 20 | 47.52 | 0 | 1.98 | 0.001 | 3.06 | 46.44 | −47.519 | 3.06 | 44.46 | |
| C. 20 (Ext) | 76.14 | 0 | 1.71 | 0.36 | 9.81 | 67.68 | −75.78 | 9.81 | 65.97 | |
| C. 21 | 40.32 | 0 | 1.62 | 5.67 | 18.27 | 18 | −34.65 | 18.27 | 16.38 | |
| C. 22 | 38.34 | 1.26 | 16.92 | 6.57 | 28.62 | 21.33 | −31.77 | 27.36 | 4.41 | |
| C. 23 | 105.3 | 5.85 | 25.65 | 79.2 | 15.39 | 42.21 | −26.1 | 9.54 | 16.56 | |
| C. 24 | 73.44 | 12.96 | 32.58 | 44.19 | 29.16 | 45.63 | −29.25 | 16.2 | 13.05 | |
| C. 25 | 48.6 | 9.72 | 56.16 | 30.6 | 19.44 | 64.43 | −18 | 9.72 | 8.27 | |
| C. 26 | 100.71 | 16.83 | 58.86 | 45.99 | 64.26 | 66.15 | −54.72 | 47.43 | 7.29 | |
| C. 27 | 101.79 | 4.77 | 29.43 | 60.93 | 28.71 | 46.35 | −40.86 | 23.94 | 16.92 | |
| Kutupalong RC | 10.17 | 13.05 | 15.75 | 7.47 | 21.33 | 10.17 | −2.7 | 8.28 | −5.58 | |
| Nayapara RC | 7.92 | 13.68 | 11.25 | 3.96 | 20.7 | 8.19 | −3.96 | 7.02 | −3.06 | |

**Table A2.** Standardized value of Per Capital Greening Area (PCGA) dataset in 2017, 2018, and 2019, by the period of observation.

| Standardized per Capital Greening Area (PCGA) Dataset | | | |
|---|---|---|---|
| Camp No. | 2017 | 2018 | 2019 |
| Camp 1W | −0.44 | −0.46 | −0.34 |
| Camp 2E | −0.82 | −0.43 | −0.33 |
| Camp 2W | −0.82 | −0.50 | −0.42 |
| Camp 3 | −0.25 | −0.50 | −0.42 |
| Camp 4 | 0.36 | −0.46 | −0.42 |
| Camp 4 Ext. | −0.56 | 0.93 | −0.23 |
| Camp 5 | −0.21 | −0.50 | −0.43 |

**Table A2.** *Cont.*

| Standardized per Capital Greening Area (PCGA) Dataset | | | |
|---|---|---|---|
| Camp No. | 2017 | 2018 | 2019 |
| Camp 6 | −0.73 | −0.50 | −0.43 |
| Camp 7 | −0.69 | −0.39 | −0.29 |
| Camp 8E | −0.23 | −0.23 | −0.09 |
| Camp 8W | −0.17 | −0.50 | −0.43 |
| Camp 9 | −0.67 | −0.43 | −0.36 |
| Camp 10 | −0.27 | −0.50 | −0.43 |
| Camp 11 | 2.48 | −0.49 | −0.41 |
| Camp 12 | 0.52 | −0.31 | −0.15 |
| Camp 13 | 0.39 | −0.47 | −0.39 |
| Camp 14 | 1.03 | −0.38 | −0.26 |
| Camp 15 | 1.07 | −0.49 | −0.42 |
| Camp 16 | −0.06 | −0.44 | −0.30 |
| Camp 17 | −0.02 | −0.07 | −0.43 |
| Camp 18 | 0.78 | −0.49 | −0.43 |
| Camp 19 | 3.94 | −0.34 | −0.28 |
| Camp 20 | −0.59 | 0.19 | −0.43 |
| Camp 20 Ext. | −0.53 | 4.32 | −0.38 |
| Camp 21 | 0.08 | −0.01 | −0.12 |
| Camp 22 | 0.08 | −0.35 | −0.23 |
| Camp 23 | −0.70 | 2.33 | 4.40 |
| Camp 24 | −0.81 | 0.12 | 0.45 |
| Camp 25 | −0.10 | 0.90 | 1.73 |
| Camp 26 | −0.51 | −0.08 | 0.32 |
| Camp 27 | 0.09 | 1.31 | 2.43 |
| Nayapara RC * | −0.82 | −0.45 | −0.33 |
| Kutupalong RC * | −0.82 | −0.30 | −0.15 |

* registered refugee camp.

**Table A3.** Spatiotemporal transition of LULC, Ukhiya-Teknaf, 2017–2018 (in km$^2$), here, Veg. = vegetative, Sett. = Settlement, Wat. = Water-body, Agri. = Agricultural land, Aqua. = Aquaculture land, Arab. = Arable land, Tidal. = Tidal Mudflat, Sand. = Sandy area, Bare. = Bare land.

| | | 2017 | | | | | | | | | |
|---|---|---|---|---|---|---|---|---|---|---|---|
| LULC Classes | | Veg. | Sett. | Wat. | Agri. | Aqua. | Arab. | Tidal. | Sand. | Bare. | C.T |
| | Veg. | 344.06 | 0.41 | 1.09 | 3.16 | 4.33 | 3.74 | 1.15 | 0.02 | 0.06 | 358.02 |
| | Sett. | 3.91 | 2.87 | 0.04 | 0.06 | 2.77 | 1.61 | 0.42 | 0.79 | 0.32 | 12.78 |
| | Wat. | 0.69 | 0.02 | 10.86 | 0 | 0.45 | 0.02 | 3.67 | 0.04 | 0 | 15.74 |
| | Agri. | 9.07 | 0.14 | 0.03 | 7.63 | 0.1 | 3.17 | 0 | 0 | 0.26 | 20.4 |
| 2018 | Aqua. | 7.82 | 2.22 | 1.12 | 0.04 | 38.91 | 2.69 | 5.78 | 0.15 | 0.09 | 58.83 |
| | Arab. | 21.49 | 6.35 | 0.05 | 1.12 | 7.51 | 35.34 | 0.1 | 0.46 | 2.07 | 74.5 |
| | Tidal. | 0.53 | 0.29 | 2.27 | 0 | 3.25 | 0.06 | 12.81 | 0.32 | 0 | 19.54 |
| | Sand. | 0.36 | 0.77 | 0.19 | 0.01 | 0.3 | 0.11 | 2.44 | 6.55 | 0.03 | 10.77 |
| | Bare. | 3.68 | 0.14 | 0 | 0.17 | 0.38 | 1.05 | 0.03 | 0.03 | 0.94 | 6.43 |
| | C.T | 391.61 | 13.21 | 15.65 | 12.2 | 58.01 | 47.79 | 26.38 | 8.38 | 3.78 | 0 |

**Table A4.** Spatiotemporal transition of LULC, Ukhiya-Teknaf, 2018–2019 (in km$^2$), here, Veg. = vegetative, Sett. = Settlement, Wat. = Water-body, Agri. = Agricultural land, Aqua. = Aquaculture land, Arab. = Arable land, Tidal. = Tidal Mudflat, Sand. = Sandy area, Bare. = Bare land.

| | **2018** | | | | | | | | | |
|---|---|---|---|---|---|---|---|---|---|---|
| **LULC Class** | **Veg.** | **Sett.** | **Wat.** | **Agri.** | **Aqua.** | **Arab.** | **Tidal.** | **Sand.** | **Bare.** | **C.T** |
| **2019** Veg. | 317.83 | 0.23 | 0.54 | 7.11 | 5.74 | 4.58 | 0.67 | 0.04 | 0.03 | 336.79 |
| Sett. | 2.31 | 7.71 | 0.07 | 0.07 | 5.43 | 8.44 | 0.9 | 1.18 | 1.47 | 27.57 |
| Wat. | 0.64 | 0.05 | 11.02 | 0 | 0.88 | 0.07 | 1.6 | 0.15 | 0 | 14.4 |
| Agri. | 5.78 | 0.04 | 0 | 6.85 | 0.04 | 1.78 | 0 | 0 | 0.02 | 14.51 |
| Aqua. | 1.36 | 0.86 | 0.04 | 0.04 | 25.02 | 2.23 | 0.82 | 0.02 | 0.05 | 30.43 |
| Arab. | 26.95 | 2.04 | 0.01 | 6 | 6.09 | 53.57 | 0.16 | 0.08 | 1.56 | 96.44 |
| Tidal. | 2.22 | 0.55 | 3.76 | 0 | 15.13 | 0.76 | 12.36 | 0.39 | 0 | 35.17 |
| Sand. | 0.1 | 0.81 | 0.29 | 0 | 0.44 | 0.49 | 3.03 | 8.91 | 0.08 | 14.15 |
| Bare. | 0.85 | 0.47 | 0 | 0.32 | 0.07 | 2.58 | 0 | 0.02 | 3.21 | 7.53 |
| C.T | 358.02 | 12.78 | 15.74 | 20.4 | 58.83 | 74.5 | 19.54 | 10.77 | 6.43 | 0 |

**Table A5.** Spatiotemporal transition of LULC, Ukhiya-Teknaf, 2017–2019 (in km$^2$), here, Veg. = vegetative, Sett. = Settlement, Wat. = Water-body, Agri. = Agricultural land, Aqua. = Aquaculture land, Arab. = Arable land, Tidal. = Tidal Mudflat, Sand. = Sandy area, Bare. = Bare land.

| | **2017** | | | | | | | | | |
|---|---|---|---|---|---|---|---|---|---|---|
| **LULC Class** | **Veg.** | **Sett.** | **Wat.** | **Agri.** | **Aqua.** | **Arab.** | **Tidal.** | **Sand.** | **Bare.** | **C.T** |
| **2019** Veg. | 322.51 | 0.4 | 0.66 | 4.39 | 5.57 | 2.44 | 0.7 | 0.04 | 0.09 | 336.79 |
| Sett. | 8.73 | 4.67 | 0.07 | 0.12 | 7.96 | 3.65 | 0.82 | 1.05 | 0.51 | 27.57 |
| Wat. | 0.41 | 0.03 | 10.16 | 0 | 0.54 | 0 | 3.16 | 0.1 | 0 | 14.4 |
| Agri. | 7.01 | 0.12 | 0.02 | 5.14 | 0.07 | 1.96 | 0 | 0 | 0.18 | 14.51 |
| Aqua. | 2.51 | 1.39 | 0.14 | 0.03 | 23.53 | 1.82 | 0.83 | 0.1 | 0.07 | 30.43 |
| Arab. | 43.03 | 4.97 | 0.07 | 2.31 | 7.44 | 36.23 | 0.1 | 0.2 | 2.08 | 96.44 |
| Tidal. | 2.1 | 0.6 | 4 | 0 | 11.88 | 0.33 | 16.09 | 0.16 | 0.01 | 35.17 |
| Sand. | 0.34 | 0.85 | 0.52 | 0.01 | 0.78 | 0.23 | 4.67 | 6.69 | 0.05 | 14.15 |
| Bare. | 4.98 | 0.18 | 0 | 0.19 | 0.23 | 1.13 | 0 | 0.03 | 0.79 | 7.53 |
| C.T. | 391.61 | 13.21 | 15.65 | 12.2 | 58.01 | 47.79 | 26.38 | 8.38 | 3.78 | 0 |

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
