# Peer review of "Analysis of Vegetative Cover Vulnerability in Rohingya Refugee Camps of Bangladesh Utilizing Landsat and Per Capita Greening Area (PCGA) Datasets"

_remotesensing, doi:10.3390/rs13234922_

Round 1

Reviewer 1 Report

This manuscript presents an analysis of vegetation cover in and around Rohingya refugee camps in Bangladesh. SVM classification is applied to dry season Landsat 8 imagery to make land cover maps. K-means unsupervised clustering was also used. A substantial decrease in vegetation cover is found over the 2017-2019 time perioud. This is obviously a high profile topic and investigation of the question is worthy of publication in Remote Sensing.

Minor to moderate English language revision is needed before publication and some issues seem to have occurred with the reference manager software, resulting in duplicated reference numbers throughout the manuscript.

Overall, I found this manuscript to provide a compelling and comprehensive study of vegetation loss in the area impacted Rohingya settlements. Ultimately, I believe it will merit publication in Remote Sensing, but some important revisions need to made beforehand. First, several of the bar chart figures are difficult to interpret and could be made much more compelling if a different plotting style were used. Second, some important aspects of the methods should be clarified – for instance, the purpose of the k-means clustering algorithm is not clear upon initial reading.

Perhaps the most important scientific point that I urge the authors to consider is the prevalence of mixed pixels in the study area. While the results of this study are sufficiently rigorous to be publishable, I strongly suggest the authors consider the extensive literature of methods that explicitly accommodate subpixel spatial mixing. In this study area, I find it highly likely that the land cover is dominated by meter-scale heterogeneity which is unlikely to be captured accurately by discrete classification methods like SVM. It might be interesting to consider approaches like spectral mixture models as a useful avenue of future work.

I recommend Major Revisions and look forward to reviewing a revised version of the manuscript soon.

Line-specific comments below:

Figure 1. The implications of this figure are not immediately apparent due to the color scheme and plotting methodology. You might try experimenting with different types of plots here to try to convey the meaning of the information more effectively.

Lines 51-78. Good job reviewing the relevant literature, but seems a bit redundant with the additional literature review in lines 99 – 135 below. You might consider merging these sections.

Lines 73-76. The meaning of “per capita greening area” should be explained here, as most readers will not be familiar with the term.

Line 106. Should this be “convolutional neural network”?

Figure 2. What is the Landsat 8 image date used here? Also, if this is using Landast 8 bands 4,3,2 it’s actually a Natural Color Composite, not a False Color Composite, since the RGB bands do correspond to visible red, green, and blue wavelengths of light.

Lines 200-215: In what way was MNF used? Why was FLAASH used instead of the standard Level-2 product?

Figure 4. I appreciate showing reflectance spectra. The contrast between the black background and dark blue Water Body spectra & text make it very challenging for the reader. Y axis bounds should be standardized so readers can easily tell differences in amplitude (e.g., “bare soil” spectral shape looks much like “settlement”, but is not obvious at first that amplitudes differ by a factor of 2).

Lines 259-270. How is “vegetative cover (Ha)” computed here? By summing #s of pixels? Are mixed pixels considered? I assume the land cover in the area is highly heterogenous and nearly all pixels are likely to be mixed…

Lines 271-295. It’s not clear how k-means was used here, and how it is related to the very informative Figure 4. K-means is unsupervised. Was k-means used in a different way, after the SVM image classification was computed?

Figure 7. Same comment as Figure 1.

Figure 8. Same comment as Figure 1.

Lines 449-607. This discussion provides a nice, compelling overview of the situation.

Author Response

We have considered the comments from the reviewers carefully and incorporated them item by item. We also made clarifications to the main body of the text, corrected grammar mistakes, misspelled words, rewrote several “clumsy and vague sentences,” and improved the quality of figures and presentations. In the list below, we have identified how we treated each comment in the revised paper.

Comments of the Author:

Reviewer #1:

General Comments:

  1. This manuscript presents an analysis of vegetation cover in and around Rohingya refugee camps in Bangladesh. SVM classification is applied to dry season Landsat 8 imagery to make land cover maps. K-means unsupervised clustering was also used. A substantial decrease in vegetation cover is found over the 2017–2019-time period. This is obviously a high-profile topic and investigation of the question is worthy of publication in Remote Sensing.

Reply: Thanks for the reviewer’s comment.

  1. Minor to moderate English language revision is needed before publication and some issues seem to have occurred with the reference manager software, resulting in duplicated reference numbers throughout the manuscript.

Reply: Thanks for the reviewer’s comment. We considered the reviewer’s suggestions and corrected grammar mistakes, misspelled words, and improved the quality of presentation in our revised manuscript.

  1. Perhaps the most important scientific point that I urge the authors to consider is the prevalence of mixed pixels in the study area. While the results of this study are sufficiently rigorous to be publishable, I strongly suggest the authors consider the extensive literature of methods that explicitly accommodate subpixel spatial mixing. In this study area, I find it highly likely that the land cover is dominated by meter-scale heterogeneity which is unlikely to be captured accurately by discrete classification methods like SVM. It might be interesting to consider approaches like spectral mixture models as a useful avenue of future work.

Reply: Thanks for the reviewer’s comment. We considered the reviewer’s suggestions and incorporate it accordingly in our revised manuscript.

We thank the reviewers for pointing at the subpixel spatial mixing issue, which we agree is important to consider in a future study. We do agree that instead of SVM the spectral mixture models seem to be a better choice, but it wouldn’t possible for us to make a side-by-side comparison of different models within the time frame of this revision. We have made a brief literature review about the use of SVM techniques for vegetation cover monitoring in line 110-124 and accounted for the subpixel mixing problem in the conclusions (see lines 630-633 in the revised paper). We suggest that approaches like spectral mixture models that consider subpixel spatial mixing should further improve our result and is a direction worthy of investigation in future work.

Line-specific comments below:

  • Figure 1. The implications of this figure are not immediately apparent due to the color scheme and plotting methodology. You might try experimenting with different types of plots here to try to convey the meaning of the information more effectively.

Reply: Thanks for the reviewer’s comment. We have tried different plot combination, and it seems we have already chosen the best fit plot for the figure-1.

  • Lines 51-78. Good job reviewing the relevant literature, but seems a bit redundant with the additional literature review in lines 99 – 135 below. You might consider merging these sections.

Reply: Thanks for the reviewer’s comment. We considered the reviewer’s suggestions and incorporate it accordingly in our revised manuscript. Please see line 102-135.

  • Lines 73-76. The meaning of “per capita greening area” should be explained here, as most readers will not be familiar with the term.

Reply: Thanks for the reviewer’s comment. We considered the reviewer’s suggestions and incorporate it accordingly in our revised manuscript. Please see line 76-80..

“Furthermore, the introduction of per Capita greening area (PCGA) dataset in this research deepens our understanding of the vegetative cover capacity changes at each of the thirty-four refugee camps. PCGA datasets are the ratio between each of the thirty-four refugee camps vegetative cover and the number of refugees in 2017, 2018, and 2019”.

  • Line 106. Should this be “convolutional neural network”?

Reply: Thanks for the reviewer’s comment. We considered the reviewer’s suggestions and incorporate it accordingly in our revised manuscript. Please see line 109.

“Yoo et al. (2019) used the convolutional neural network (CNN) for climate zone classification”.

  • Figure 2. What is the Landsat 8 image date used here? Also, if this is using Landast 8 bands 4,3,2 it’s actually a Natural Color Composite, not a False Color Composite, since the RGB bands do correspond to visible red, green, and blue wavelengths of light.

Reply: Thanks for the reviewer’s comment. We considered the reviewer’s suggestions and incorporate it accordingly in our revised manuscript. Please see line 150-159.

“Fig. 2 A schematic framework of Ukhiya-Teknaf, study area. Figure 2 (a) shows Bangladesh bordered by India, Myanmar, and the Bay of Bengal. Figure 2 (b) displays the Ukhiya-Teknaf sub-district, represented as red color, bounded by the Ramu sub-district on the north, Arakan state of Myanmar & Naikhongchori sub-district of Bangladesh on the east, and the Bay of Bengal on the west and south corner. These two-adjoining sub-district are the focus of our analysis of temporal vegetative cover monitoring before and after the Rohingya refugee crisis and eventually identify the influencing factors behind the declining of vegetative cover. Figure 2 (c) shows a natural color composite Landsat 8 image of the study area, Ukhiya-Teknaf, with a band combination of 4, 3, and 2 shows all the existing Rohingya refugee camps dated 4 February, 2019). These camps are the focus of our identification and analysis of refugee camps with highly vulnerable vegetative cover”.

  • Lines 200-215: In what way was MNF used? Why was FLAASH used instead of the standard Level-2 product?

Reply: Thanks for the reviewer’s comment. We considered the reviewer’s suggestions and incorporate it accordingly in our revised manuscript. Please see line 207-211.

“The minimum noise fraction (MNF) wizard used in this study in order to segregated noise from the data, and to reduce the computational requirements for the subsequent processing”.

  • Lines 259-270. How is “vegetative cover (Ha)” computed here? By summing #s of pixels? Are mixed pixels considered? I assume the land cover in the area is highly heterogenous and nearly all pixels are likely to be mixed…

Reply: Thanks for the reviewer’s comment. We considered the reviewer’s suggestions and incorporate it accordingly in our revised manuscript.

Initially, we have produced vegetative land cover using the professional familiarity of the study area, field survey data, and observations of Google Earth historical images of 13 February 2017, and 13 February 2018, and photo interpretation to identify and confirm the diverse land cover features. The training areas were produced utilizing polygon vectors based on the spectral reflectance wavelength, presented in Fig. 4 and Table 2. The vegetative cover (ha) was computed by summing the pixels with spectral reflectance values a like.

  • Lines 271-295: It’s not clear how k-means was used here, and how it is related to the very informative Figure 4. K-means is unsupervised. Was k-means used in a different way, after the SVM image classification was computed?

Reply: Thanks for the reviewer’s comment. We considered the reviewer’s suggestions and incorporate it accordingly in our revised manuscript.

The k-means classification technique permits a deeper understanding of the vegetative cover capacity changes due to the sudden influx of refugees at each of the thirty-four refugee camps, and eventually identification of the refugee camps with the highly vulnerable vegetative cover over the study period. K-means classification identifies observations that are alike for categorization.

Firstly, we have calculated the vegetative cover of each of the 34 refugee camps in the year of 2017, 2018, and 2019, respectively using SVM classification techniques. Secondly, we have created the PCGA dataset and the PCGA dataset is used here to calculate the refugee population-wise vegetative cover in each of the thirty-four refugee camps in sequence. Then we have standardized the dataset. The K-means classification and PCGA dataset is used in this research in order to identify the highly vulnerable vegetative cover refugee camps.

  • Figure 7. Same comment as Figure 1.

Reply: Thanks for the reviewer’s comment. We have tried different plot combination, and it seems we have already chosen the best fit plot for the figure-1.

  • Figure 8. Same comment as Figure 1.

Reply: Thanks for the reviewer’s comment. We have tried different plot combination, and it seems we have already chosen the best fit plot for the figure-1.

  • Lines 449-607. This discussion provides a nice, compelling overview of the situation.

Reply: Thanks for the reviewer’s comment.

Reviewer 2 Report

The manuscript is well written. However, there are a number of issues that can be improved.

The authors set one of ultimate goals of the study to identify critical factors behind the declining the vegetative cover. I expected a thorough investigation of different possible causes of the vegetation cover loss. While the authors linked the vegetative cover loss to the number of people in the refugee camps. I agree that the number of people is important, but it is a "macro" factor and it looks quite obvious. In fact, different activities of the people may cause declining of vegetative cover, e.g. clearing land for building new camps, expansion of agriculture land, fuelwood collection, etc. 

I suggest the authors to modify the goal slightly to make it closer to the real results of the study.

There are many similar technical mistakes to be corrected through the text:

  • In the introduction the reference numbers are duplicated;
  • When the authors refer to a publication by an author name and a year then a a number reference should follow immediately to make clear which name and year corresponds to which number;
  • "Per Capital Greening Area" should be "Per Capita Greening Area".

L80: ... refugee camps with highly vulnerable vegetation ....

L109: SVM to defined here.

L178: There are two registered ....

L182: Camp 15

Table 2: ... temporary fallow land

Table 5: Not clear what do the "Vegetative" and "Non-vegetative" classes are.  To which of those the agricultural land belongs? Why Settlements are not with non-vegetative?

L376: Table A1 shows....

L378: Camp 4

Fig.8 upper: The marks at the horizontal axis are very small.

L421: ...the cluster must be set before....

Table 6: Not clear how the vulnerability is defined, how do the authors compare the vulnerability of the clusters? Why for the cluster 2 the mean values increase in 2017-2018 (the vulnerability decreases? More refugees - less vulnerability?)?

L450: The title of 5.1 is too long.

L582: Fig. 17 caption is missing or too long.

Author Response

We have considered the comments from the reviewers carefully and incorporated them item by item. We also made clarifications to the main body of the text, corrected grammar mistakes, misspelled words, rewrote several “clumsy and vague sentences,” and improved the quality of figures and presentations. In the list below, we have identified how we treated each comment in the revised paper.

Comments of the Author:

Reviewer #2:

General Comments:

  1. The authors set one of ultimate goals of the study to identify critical factors behind the declining the vegetative cover. I expected a thorough investigation of different possible causes of the vegetation cover loss. While the authors linked the vegetative cover loss to the number of people in the refugee camps. I agree that the number of people is important, but it is a "macro" factor and it looks quite obvious. In fact, different activities of the people may cause declining of vegetative cover, e.g., clearing land for building new camps, expansion of agriculture land, fuelwood collection, etc. I suggest the authors to modify the goal slightly to make it closer to the real results of the study.

Reply: Thanks for the reviewer’s comment. We have considered the reviewer’s suggestions and incorporate it in our revised manuscript. Please see line 688-691.

“Besides the rapid influx of Rohingya`s, different activities such as clearing land for building news camps (reference), expansion of agricultural land (reference), cutting down the forest for fuelwood collections (reference), and so on are the critical factors behind the declining of vegetative cover in Ukhiya-Teknaf. The refugee demands 750,000 kg of fuelwood each day, and to meet this substantial demand, they razed down in and surrounding protected forest [2]. In addition, nearly a million refugees have accommodated a total area of only 2510.01 hectares of hilly land. As a consequence, rampant hill wiping out the herbaceous layer may trigger landslides during the rainy season. Landslides are a frequent natural disaster in the hilly mountain areas in Bangladesh and cost many lives each year [14]. A gigantic landslide in the campsite might prompt a more significant humanitarian situation. Though the government of Bangladesh strictly prohibited the expansion of the refugee camp further to protect the reserve forest, many refugees are camping in the deep forest and blocking the elephant corridors [38]. Severe deforestation is resulting at local and regional levels to meet the demand within the refugee camps. The international organization for migration estimated 3,000 out of 43,000 acres’ forest land in the hilly district of Cox’s Bazar is destroyed by the refugees since the crisis broke out, resulting in a rapid increase of refugee settlement and bare areas [9]”.

  1. In the introduction the reference numbers are duplicated. When the authors refer to a publication by an author name and a year then a a number reference should follow immediately to make clear which name and year corresponds to which number

Reply: Thanks for the reviewer’s comment. We have considered the reviewer’s suggestions and incorporate it in our revised manuscript.

  1. "Per Capital Greening Area" should be "Per Capita Greening Area"

Reply: Thanks for the reviewer’s comment. We have considered the reviewer’s suggestions and incorporate it in our revised manuscript.

Line-specific comments below:

L80: ... refugee camps with highly vulnerable vegetation ....

Reply: Thanks for the reviewer’s comment. We have considered the reviewer’s suggestions and incorporate it in our revised manuscript.

L109: SVM to defined here.

Reply: Thanks for the reviewer’s comment. We have considered the reviewer’s suggestions and incorporate it in our revised manuscript.

L178: There are two registered ....

Reply: Thanks for the reviewer’s comment. We have considered the reviewer’s suggestions and incorporate it in our revised manuscript.

L182: Camp 15

Reply: Thanks for the reviewer’s comment. We have considered the reviewer’s suggestions and incorporate it in our revised manuscript.

Table 2: ... temporary fallow land

Reply: Thanks for the reviewer’s comment. We have considered the reviewer’s suggestions and incorporate it in our revised manuscript.

Table 5: Not clear what do the "Vegetative" and "Non-vegetative" classes are.  To which of those the agricultural land belongs? Why Settlements are not with non-vegetative?

Reply: Thanks for the reviewer’s comment. We have considered the reviewer’s suggestions and incorporate it in our revised manuscript.

In this part of the research, the main focus was to observe the transformation of vegetative (mainly forest land) cover, settlement area, and non-vegetative (includes; waterbody, agricultural land, aquaculture land, arable land, tidal mudflat, and sandy area) area over year in the study area.

In this case we incorporate agricultural land and other land classes into non-vegetative group. Besides, we took settlement area as an individual class in order to monitor total net vegetative, non-vegetative, and settlement cover changes in all thirty-four camps. And the results shows that the net vegetative, non-vegetative, and settlement cover changes in all thirty-four camps are -1502.56 hectares, +760.89 hectares, and +729.99 hectares, respectively.

Please see L528-532

“The conversion matrix of land cover in 2017-2019 suggests that vegetative to non-vegetative (i.e. waterbody, agricultural land, aquaculture land, arable land, tidal mudflat, and sandy area) land cover increased rapidly, accounting for 956 hectares; additionally, vegetative to settlement and non-vegetative to settlement conversion area was 546 hectares and 209 hectares”.

L376: Table A1 shows....

Reply: Thanks for the reviewer’s comment. We have considered the reviewer’s suggestions and incorporate it in our revised manuscript.

L378: Camp 4

Reply: Thanks for the reviewer’s comment. We have considered the reviewer’s suggestions and incorporate it in our revised manuscript.

Fig.8 upper: The marks at the horizontal axis are very small.

Reply: Thanks for the reviewer’s comment. We have considered the reviewer’s suggestions and incorporate it in our revised manuscript.

L421: ...the cluster must be set before....

Reply: Thanks for the reviewer’s comment. We have considered the reviewer’s suggestions and incorporate it in our revised manuscript.

Table 6: Not clear how the vulnerability is defined, how do the authors compare the vulnerability of the clusters? Why for the cluster 2 the mean values increase in 2017-2018 (the vulnerability decreases? More refugees - less vulnerability?)?

Reply: Thanks for the reviewer’s comment.

The PCGA dataset is used to calculate the refugee population-wise vegetative cover in each of the thirty-four refugee camps in sequence. The equation used in this research to compute the PCGA is as follows:

PCGA =

The K-means classification and PCGA dataset is used in this research in order to identify the highly vulnerable vegetative cover refugee camps. In this research, cluster-2 is identified as less vulnerable cluster group based of their average mean value. The average mean values of cluster 2 is depended on the Standardize PCGA value of each of the 4 refugee camps includes; C-20 (ext.), C-23, C-25, and 27. The higher the mean values the lower the vulnerability. Here the vulnerability not only dependent on the changing number of refugees between 2017 and 2019 but also dependent on the changes of vegetative cover area in each camp over the year. Table A2 shows the Standardize value of Per Capita Greening Area (PCGA) dataset in 2017, 2018, and 2019, by the period of observation. Please see Line 892-895.

Table A 1 Standardize value of Per Capita Greening Area (PCGA) dataset in 2017, 2018, and 2019, by the period of observation

Standardize Per Capita Greening Area (PCGA) dataset

2017

2018

2019

Camp 1W

-0.44

-0.46

-0.34

Camp 2E

-0.82

-0.43

-0.33

Camp 2W

-0.82

-0.50

-0.42

Camp 3

-0.25

-0.50

-0.42

Camp 4

0.36

-0.46

-0.42

Camp 4 Ext.

-0.56

0.93

-0.23

Camp 5

-0.21

-0.50

-0.43

Camp 6

-0.73

-0.50

-0.43

Camp 7

-0.69

-0.39

-0.29

Camp 8E

-0.23

-0.23

-0.09

Camp 8W

-0.17

-0.50

-0.43

Camp 9

-0.67

-0.43

-0.36

Camp 10

-0.27

-0.50

-0.43

Camp 11

2.48

-0.49

-0.41

Camp 12

0.52

-0.31

-0.15

Camp 13

0.39

-0.47

-0.39

Camp 14

1.03

-0.38

-0.26

Camp 15

1.07

-0.49

-0.42

Camp 16

-0.06

-0.44

-0.30

Camp 17

-0.02

-0.07

-0.43

Camp 18

0.78

-0.49

-0.43

Camp 19

3.94

-0.34

-0.28

Camp 20

-0.59

0.19

-0.43

Camp 20 Ext.

-0.53

4.32

-0.38

Camp 21

0.08

-0.01

-0.12

Camp 22

0.08

-0.35

-0.23

Camp 23

-0.70

2.33

4.40

Camp 24

-0.81

0.12

0.45

Camp 25

-0.10

0.90

1.73

Camp 26

-0.51

-0.08

0.32

Camp 27

0.09

1.31

2.43

Nayapara RC*

-0.82

-0.45

-0.33

Kutupalong RC*

-0.82

-0.30

-0.15

L450: The title of 5.1 is too long.

Reply: Thanks for the reviewer’s comment. We have considered the reviewer’s suggestions and incorporate it in our revised manuscript.

L582: Fig. 17 caption is missing or too long.

Reply: Thanks for the reviewer’s comment. We have considered the reviewer’s suggestions and incorporate it in our revised manuscript.

Round 2

Reviewer 1 Report

I appreciate the effort the authors have taken to address the review concerns and am satisfied with the paper being published in current form.